# ARLArena: A Unified Framework for Stable Agentic Reinforcement Learning

Xiaoxuan Wang [* 1]  Han Zhang [* 1]  Haixin Wang [* 1]  Yidan Shi [† 1]  Ruoyan Li [† 1]  Kaiqiao Han [† 1]  Chenyi Tong [2]
Haoran Deng [1]  Alexander K Taylor [1]  Renliang Sun [1]  Yanqiao Zhu [1]  Jason Cong [1]  Yizhou Sun [1]  Wei Wang [1]

## Abstract

Agentic reinforcement learning (ARL) has rapidly gained attention as a promising paradigm for training agents to solve complex, multi-step interactive tasks. Despite encouraging early results, ARL remains highly unstable, often leading to training collapse. This instability limits its scalability to larger environments and longer interaction horizons, and constrains systematic exploration of algorithmic design choices. In this paper, we first propose **ARLArena**, a stable training recipe and systematic analysis framework that examines training stability in a controlled and reproducible setting. ARLArena first constructs a clean and standardized testbed. Then, we decompose policy gradient into four core design dimensions and assess the performance and stability of each dimension. Through this fine-grained analysis, we propose **SAMPO**, a stable agentic policy optimization method designed to mitigate the dominant sources of instability in ARL. Empirically, SAMPO achieves consistently stable training and strong performance across diverse agentic tasks. Our code is open-sourced at: `https://github.com/WillDreamer/ARL-Arena.git`

## 1. Introduction

Large language models (LLMs) have been increasingly deployed as autonomous agents for complex, multi-step interactive tasks spanning web navigation (Zhou et al., 2024; Wang et al., 2026), embodied environments (Shridhar et al., 2020), games (Xi et al., 2024), and deep research (Jin et al., 2025; Guan et al., 2025). These tasks demand planning,

tool use, and long-horizon decision-making, necessitating training objectives that capture multi-turn interactions. Reinforcement learning (RL) offers a principled post-training framework for this purpose, building on its success in static reasoning tasks (*e.g.*, DeepSeek-R1 (Guo et al., 2025), OpenAI o1 (Jaech et al., 2024)), and early results in the agentic setting are promising (Jin et al., 2025; Cheng et al., 2025; Xi et al., 2024).

However, agentic RL (ARL) training remains highly unstable and prone to collapse (Xi et al., 2025). This instability arises from the interactive, multi-turn nature of agentic environments, which introduce compounding challenges such as invalid actions, sparse rewards, long-horizon credit assignment, and non-stationary agent–environment dynamics (Wang et al., 2025b; Xu et al., 2026). Small deviations in early decisions can cascade across turns, causing distribution shifts that amplify credit-assignment noise and produce degenerate rollouts (Xia et al., 2026; Xie et al., 2026). Consequently, ARL outcomes are difficult to reproduce across runs and environments, and scaling to longer horizons or more complex interaction spaces remains severely limited (Abdulhai et al., 2023; Xi et al., 2025). These challenges underscore the need for stable and scalable training solutions for ARL.

This paper addresses this gap by introducing **ARLArena**, a stable training recipe and systematic analysis framework for agentic reinforcement learning. We first construct a clean, standardized testbed through format correction, behavior cloning initialization, and KL-based regularization, establishing reliable baseline performance. We then decompose policy-gradient–based RL into four orthogonal design dimensions and evaluate the effectiveness and stability of each across diverse agentic tasks. Each dimension is examined in isolation using representative policy optimization (PO) methods; for methods that exhibit training collapse, we further diagnose the underlying failure modes and develop targeted stabilization strategies.

This systematic analysis yields three key findings: (1) tolerant clipping induces training collapse, whereas sequence-level clipping ensures stable improvement; (2) incorporating environment-level information into advantage design improves both stability and performance; and (3) dynamic

---

*Equal contribution †Contributed equally as second authors [1]Department of Computer Science, University of California, Los Angeles, California, USA [2]University of Wisconsin–Madison, Madison, Wisconsin, USA. Correspondence to: Xiaoxuan Wang <xw27@ucla.edu>, Haixin Wang <whx@ucla.edu>.

*Proceedings of the 43rd International Conference on Machine Learning*, Seoul, South Korea. PMLR 306, 2026. Copyright 2026 by the author(s).

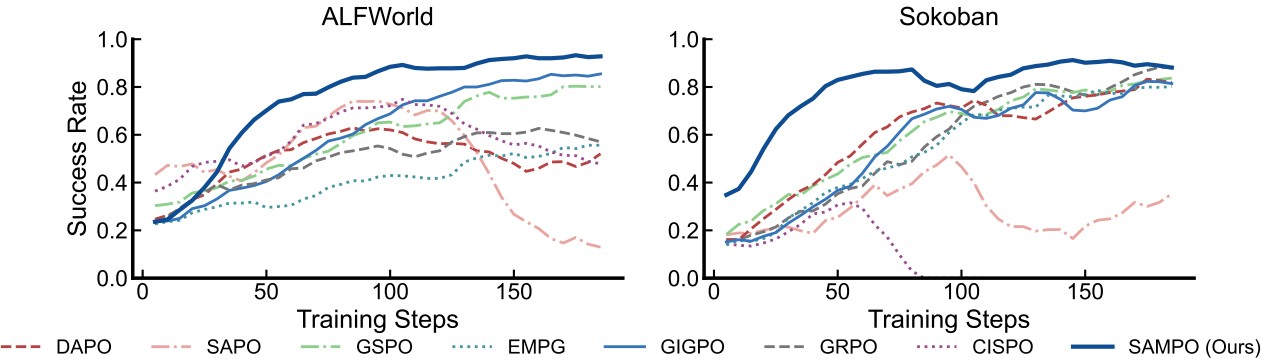

*Figure 1.* Training curves on ALFWorld (left) and Sokoban (right). Our proposed SAMPO achieves the highest success rates on both environments with stable, monotonic improvement throughout training, while baseline methods exhibit varying degrees of instability. These results demonstrate that principled integration of sequence-level clipping, advantage design, and dynamic filtering, as combined in SAMPO, is critical for both training stability and final performance in multi-turn ARL.

sampling combined with fine-grained advantage design further benefits ARL training. Motivated by these insights, we propose **S**table **A**gentic **M**ulti-turn **P**olicy **O**ptimization (**SAMPO**), a unified PO method that directly addresses the dominant sources of instability identified in our analysis. SAMPO consistently improves training stability and performance, achieving an average **25.2%** improvement over the GRPO baseline. We additionally study the impact of off-policy staleness in agentic environments and conduct comparative evaluations against proprietary models, demonstrating the robustness and generality of ours.

In summary, our contributions are: (i) a unifying policy gradient perspective and four-dimensional categorization of PO methods for ARL; (ii) a standardized, reproducible testbed and diagnostic methodology for multi-turn ARL stability; (iii) principled, task-robust findings and remedies for common collapse modes; and (iv) SAMPO, a new PO method that achieves both reliable training and strong final performance. We hope this study provides a foundation for more principled progress in agentic post-training.

## 2. Problem Formulation

### 2.1. Policy Gradient for Agentic RL

During RL optimization for LLMs, the policy $\pi_\theta$ generates a response trajectory $y = (y_0, \ldots, y_T)$ conditioned on a prompt $x$, which is subsequently used for policy updates (Ouyang et al., 2022). Following PPO-style optimization (Schulman et al., 2017), trajectories collected under a behavior policy $\pi_{\theta_{\text{old}}}$ are used to update the current policy $\pi_\theta$. The corresponding policy gradient can be written as:

$$\nabla_\theta \mathcal{L}(\theta) = \mathbb{E}_{y \sim \pi_{\theta_{\text{old}}}} \left[ \sum_{t=0}^{T} w_t(y) \nabla_\theta \log \pi_\theta(y_t \mid x, y_{<t}) \, A(x, y) \right], \quad (1)$$

where the importance sampling weight is given by:

$$w_t(y) = \frac{P_\theta(y_t \mid x, y_{<t})}{P_{\theta_{\text{old}}}(y_t \mid x, y_{<t})} = \frac{\pi_\theta(y_t \mid x, y_{<t})}{\pi_{\theta_{\text{old}}}(y_t \mid x, y_{<t})}. \quad (2)$$

Here, $A(x, y)$ represents the advantage.

**Agentic RL.** An agent interacts with the environment over $K$ turns, forming a long-horizon decision-making process (Wei et al., 2026; Luo et al., 2026). At each turn, the policy conditions on the accumulated history to generate a response, from which an action is extracted and executed to transition the environment state.

The initial user prompt is $x^{(1)}$. At turn $k \in \{1, \ldots, K\}$, the policy generates a response $y^{(k)} \sim \pi_\theta(\cdot \mid x^{(k)})$. Given the environment state $s^{(k)}$, actions $a^{(k)}$ are extracted from $y^{(k)}$, and the environment transitions to the next state $s^{(k+1)}$ according to an update function $f$: $s^{(k+1)} = f(a^{(k)}, s^{(k)})$, where $f(\cdot)$ is the state transition function that incorporates tool calls, environment observations, or retrieved information. The user prompt for turn $k + 1$, denoted $x^{(k+1)}$, is constructed from the updated state $s^{(k+1)}$. Finally, the complete multi-turn interaction trajectory is defined as $\tau = (x^{(1)}, y^{(1)}, x^{(2)}, y^{(2)}, \ldots, x^{(K)}, y^{(K)})$.

In the multi-turn agent–environment setting described above, we decompose a $K$-turn trajectory into single-turn updates. This yields the following policy gradient formulation for agentic LLM interaction:

$$\nabla_\theta \mathcal{L}(\theta) = \mathbb{E}_{\tau \sim \pi_{\theta_{\text{old}}}} \Big[ \sum_{k=1}^{K} \sum_{t=0}^{T_k} \underbrace{w_t(y^{(k)})}_{\text{IS}}$$
$$\underbrace{\nabla_\theta \log \pi_\theta\left(y_t^{(k)} \mid x^{(k)}, y_{<t}^{(k)}\right)}_{\text{Log prob}} \underbrace{A(x^{(k)}, y^{(k)})}_{\text{Advantage}} \Big]. \quad (3)$$

### 2.2. Policy Gradient Decomposition Dimensions

According to Equation 3, the policy gradient formulation for agentic LLMs can be decomposed into four key re-

| Method | Loss Objective | Advantage ($A_i$) | IS ($w_t$) Clipping | | Dynamical Sampling |
|---|---|---|---|---|---|
| | | | **Adv < 0** | **Adv > 0** | |
| GRPO | $\frac{1}{\sum_{i=1}^{G} T_i} \sum_{i=1}^{G} \sum_{t=0}^{T_i-1} \min\big(w_t A_i,\ \mathrm{clip}(w_t, 1\pm\varepsilon)A_i\big)$ | $\frac{r_i - \mathrm{mean}(r_i)}{\mathrm{std}(r_i)}$ | $\begin{cases} 1-\varepsilon, & w_t < 1-\varepsilon, \\ w_t, & \text{otherwise.} \end{cases}$ | $\begin{cases} 1+\varepsilon & w_t > 1+\varepsilon, \\ w_t, & \text{otherwise.} \end{cases}$ | ✗ |
| GRPO$_{\text{ST}}$ | $\frac{1}{G} \sum_{i=1}^{G} \frac{1}{T_i} \sum_{t=0}^{T_i-1} \min\big(w_t A_i, \mathrm{clip}(w_t, 1\pm\varepsilon)A_i\big)$ | $\frac{r_i - \mathrm{mean}(r_i)}{\mathrm{std}(r_i)}$ | $\begin{cases} 1-\varepsilon, & w_t < 1-\varepsilon, \\ w_t, & \text{otherwise.} \end{cases}$ | $\begin{cases} 1+\varepsilon & w_t > 1+\varepsilon, \\ w_t, & \text{otherwise.} \end{cases}$ | ✗ |
| GRPO$_{\text{SM}}$ | $\frac{1}{\sum_{i=1}^{G} T_i} \sum_{i=1}^{G} \sum_{t=0}^{T_i-1} M_i \min\big(w_t A_i,\ \mathrm{clip}(w_t, 1\pm\varepsilon)A_i\big)$ $M_i = \mathbf{1}\left[A_i \geq 0 \text{ or } \frac{1}{|T_i|} \sum_{t=0}^{|T_i|-1} \log \frac{\pi_{\theta_{\text{old}}}(y_t|x,y_{<t})}{\pi_\theta(y_t|x,y_{<t})} \leq \delta \right]$ | $\frac{r_i - \mathrm{mean}(r_i)}{\mathrm{std}(r_i)}$ | $\begin{cases} 1-\varepsilon, & w_t < 1-\varepsilon, \\ w_t, & \text{otherwise.} \end{cases}$ | $\begin{cases} 1+\varepsilon & w_t > 1+\varepsilon, \\ w_t, & \text{otherwise.} \end{cases}$ | ✗ |
| SAPO | $\frac{1}{\sum_{i=1}^{G} T_i} \sum_{i=1}^{G} \sum_{t=0}^{T_i-1} f_{i,t}(w_t)A_i$ | $\frac{r_i - \mathrm{mean}(r_i)}{\mathrm{std}(r_i)}$ | $\sigma(\tau_{\text{neg}}(w_t - 1)) \cdot \frac{4}{\tau_{\text{neg}}}$ | $\sigma(\tau_{\text{pos}}(w_t - 1)) \cdot \frac{4}{\tau_{\text{pos}}}$ | ✗ |
| CISPO | $\frac{1}{\sum_{i=1}^{G} T_i} \sum_{i=1}^{G} \sum_{t=0}^{T_i-1} \mathrm{sg}(w_t) A_i \log \pi_\theta$ | $\frac{r_i - \mathrm{mean}(r_i)}{\mathrm{std}(r_i)}$ | $\begin{cases} 1-\varepsilon_{\text{low}}, & w_t < 1-\varepsilon_{\text{low}}, \\ \mathrm{sg}(w_t), & \text{otherwise.} \end{cases}$ | $\begin{cases} 1+\varepsilon_{\text{high}}, & w_t > 1+\varepsilon_{\text{high}}, \\ \mathrm{sg}(w_t), & \text{otherwise.} \end{cases}$ | ✗ |
| GSPO | $\frac{1}{\sum_{i=1}^{G} T_i} \sum_{i=1}^{G} \sum_{t=0}^{T_i-1} \min\big(s_i A_i,\ \mathrm{clip}(s_i, 1\pm\varepsilon)A_i\big)$ $s_i = \exp\Big(\frac{1}{|T_i|} \sum_{t=0}^{|T_i|-1} \log \frac{\pi_\theta(y_t \mid x, y_{<t})}{\pi_{\theta_{\text{old}}}(y_t \mid x, y_{<t})}\Big)$ | $\frac{r_i - \mathrm{mean}(r_i)}{\mathrm{std}(r_i)}$ | $\begin{cases} 1-\varepsilon, & s_i < 1-\varepsilon, \\ s_i, & \text{otherwise.} \end{cases}$ | $\begin{cases} 1+\varepsilon, & s_i > 1+\varepsilon, \\ s_i, & \text{otherwise.} \end{cases}$ | ✗ |
| GIGPO | $\frac{1}{\sum_{i=1}^{G} T_i} \sum_{i=1}^{G} \sum_{t=0}^{T_i-1} \min\big(w_t A'_{i,k}, \mathrm{clip}(w_t, 1\pm\varepsilon)A'_{i,k}\big)$ | $A_i + \omega \cdot A_{\text{step}}(\hat{y}_{i,k})$ | $\begin{cases} 1-\varepsilon, & w_t < 1-\varepsilon, \\ w_t, & \text{otherwise.} \end{cases}$ | $\begin{cases} 1+\varepsilon, & w_t > 1+\varepsilon, \\ w_t, & \text{otherwise.} \end{cases}$ | ✗ |
| EMPG | $\frac{1}{\sum_{i=1}^{G} T_i} \sum_{i=1}^{G} \sum_{t=0}^{T_i-1} \min\big(w_t A'_i, \mathrm{clip}(w_t, 1\pm\varepsilon)A'_i\big)$ | $g(H_k)A_i + \zeta f(H_{k+1})$ | $\begin{cases} 1-\varepsilon, & w_t < 1-\varepsilon, \\ w_t, & \text{otherwise.} \end{cases}$ | $\begin{cases} 1+\varepsilon, & w_t > 1+\varepsilon, \\ w_t, & \text{otherwise.} \end{cases}$ | ✗ |
| DAPO | $\frac{1}{\sum_{i=1}^{G} T_i} \sum_{i=1}^{G} \sum_{t=0}^{T_i-1} \min\big(w_t A_i,\ \mathrm{clip}(w_t, 1\pm\varepsilon)A_i\big)$ | $\frac{r_i - \mathrm{mean}(r_i)}{\mathrm{std}(r_i)}$ | $\begin{cases} 1-\varepsilon_{\text{low}}, & w_t < 1-\varepsilon_{\text{low}}, \\ w_t, & \text{otherwise.} \end{cases}$ | $\begin{cases} 1+\varepsilon_{\text{high}}, & w_t > 1+\varepsilon_{\text{high}}, \\ w_t, & \text{otherwise.} \end{cases}$ | ✓ |

*Table 1.* A summary of policy optimization methods studied in ARLArena, decomposed along four design dimensions: loss objective formulation, advantage ($A_i$), importance sampling (IS) clipping, and dynamic sampling. Colored entries highlight distinctive design choices: purple denotes modified loss aggregation (seq-mean-token-mean), violet indicates alternative IS clipping strategies (tolerant or sequence-level), and green marks novel advantage designs. The importance sampling weight is $w_t = \pi_\theta(y_t \mid x, y_{<t})/\pi_{\theta_{\text{old}}}(y_t \mid x, y_{<t})$, and $\mathrm{sg}(\cdot)$ denotes the stop-gradient operator.

search dimensions: Loss Aggregation, Importance Sampling (IS) clipping, Trajectory Filtering and Resampling, and Advantage Design. To study each dimension in isolation, we analyze the batch-level loss objective without loss of generality. We summarize mainstream PO algorithms across the different design dimensions of the policy gradient in Table 1.

**Loss Aggregation.** In practice, we approximate the loss objective using different loss aggregation schemes.

$$\mathcal{L}(\theta) = \mathbb{E}_{y^{(i)} \sim \pi_{\theta_{\text{old}}}} \left[ \mathbb{E}_t \left[ \ell_{i,t}(\theta) \right] \right]$$

$$\triangleq \frac{1}{N} \sum_{i=1}^{N} \frac{1}{T_i} \sum_{t=0}^{T_i-1} \ell_{i,t}(\theta) \quad \text{(seq-mean-token-mean)} \quad (4)$$

$$\triangleq \frac{1}{\sum_{i=1}^{N} T_i} \sum_{i=1}^{N} \sum_{t=0}^{T_i-1} \ell_{i,t}(\theta) \quad \text{(token-mean)}, \quad (5)$$

where $\ell_{i,t}(\theta) := \min\big(w_{i,t}(\theta) A_i,\ \mathrm{clip}\big(w_{i,t}(\theta), 1-\varepsilon, 1+\varepsilon\big) A_i\big)$. $N$ denotes the total number of decomposed turns over trajectories. $A_i$ denotes the advantage of sequence $y^{(i)}$, and $w_{i,t}(\theta)$ is the importance sampling ratio at token $t$ of sequence $y^{(i)}$. Seq-mean-token-mean weights each token by the inverse of its trajectory length, biasing optimization toward shorter trajectories and potentially introducing response-level length bias. Token-mean assigns equal

weight to all unmasked tokens in the batch. Additional aggregation strategies are provided in the Appendix A.1.

**IS Clipping.** Clipping methods constrain the magnitude of policy updates by limiting the change in action probabilities relative to the old policy. By constraining the deviation between the new and old policies within a bounded range, clipping mitigates performance degradation and instability caused by excessively large policy updates. The loss objective is formulated as follows:

$$\mathcal{L}(\theta) = \frac{1}{\sum_{i=1}^{N} T_i} \sum_{i=1}^{N} \sum_{t=0}^{T_i-1} \min\Big(w_{i,t}(\theta) A_i,\ \mathrm{clip}\big(w_{i,t}(\theta), 1\pm\varepsilon\big) A_i\Big). \quad (6)$$

Within the GRPO (Guo et al., 2025) framework, several clipping variants are considered, including CISPO (Chen et al., 2025), SAPO (Gao et al., 2025), and GSPO (Zheng et al., 2025). CISPO employs a stop-gradient mechanism to avoid hard clipping of out-of-bounds tokens while preserving their gradient information. SAPO adopts a soft-clipping strategy, in which excessively large ratios are smoothly attenuated rather than truncated. GSPO performs clipping by using the sequence-level importance ratio as the clipping criterion. Detailed formulations of these variants

| Algorithm | Strategy | Task Score | Success Rate |
|---|---|---|---|
| GRPO | + Behavior Cloning | + 2.56 | + 20.71 |
| | + $\mathcal{R}_{\text{format}}$ | + 0.49 | + 7.34 |
| | + KL $k_3(x)$ | + 0.95 | + 18.10 |
| GSPO | $\epsilon : e^{-2} \rightarrow e^{-3}$ | + 0.70 | + 3.36 |
| | $\epsilon : e^{-3} \rightarrow e^{-4}$ | − 1.16 | − 9.88 |
| DAPO | Max_try: $2 \rightarrow 3$ | + 0.59 | + 22.15 |
| SAPO | Temperature: $1 \rightarrow 2$ | − 1.20 | − 9.85 |
| | Temperature: $2 \rightarrow 3$ | − 0.70 | − 9.20 |

*Table 2.* Incremental stabilization strategies for constructing a standardized testbed on ALFWorld, evaluated using GRPO as the base policy optimizer. Each row adds one stabilization technique or adjusts a method-specific hyperparameter. Task Score and Success Rate report the absolute improvement (+) or degradation (−) relative to the preceding configuration.

are provided in Table 1 and Appendix A.2.

**Trajectory Filtering and Resampling.** Dynamic sampling addresses inefficiency caused by zero-gradient trajectories in long-horizon agent training (Yu et al., 2025a).

$$\mathcal{L}(\theta) = \frac{1}{\sum_{i=1}^{N} T_i} \sum_{i=1}^{N} \sum_{t=0}^{T_i - 1} \min\left( w_{i,t}(\theta) A_i, \ \text{clip}(w_{i,t}(\theta), 1 \pm \varepsilon) A_i \right),$$
$$\text{s.t.} \quad 0 < \left| \left\{ y^{(i)} \ \middle| \ \text{is\_equivalent}(a, y^{(i)}) \right\} \right| < G. \tag{7}$$

Here, $a$ denotes the ground-truth task completion target, and equivalence is determined by whether the agent successfully completes the task. It adaptively filters out trajectories whose sampled output groups receive identical rewards (e.g., all correct or all incorrect) and resamples additional trajectories to increase the proportion of samples with informative gradient signals.

**Advantage Design.** Multi-turn agentic reinforcement learning introduces additional interaction steps and explicit agent–environment state transitions, which motivates specialized advantage designs. GiGPO (Feng et al., 2025) defines advantages at the state level by grouping actions conditioned on the same preceding environment state and assigning them a shared relative advantage. EMPG (Wang et al., 2025a) augments the advantage function with an entropy-dependent term, which modulates the learning signal at each turn to better account for uncertainty across interaction steps. Detailed formulations of these variants are provided in Table 1 and Appendix A.3.

## 3. Experimental Setup

### 3.1. Standardized Testbed

A primary challenge is constructing a fair and effective testbed for comparing different algorithms. To address this issue, we progressively apply a sequence of stabilization strategies shown in Table 2. Specifically, we start with be-

havior cloning, followed by format penalty enforcement and KL regularization when necessary, and finally PO-specific hyperparameter tuning. This process yields a standardized and stable testbed that provides a solid foundation for systematically comparing different PO strategies.

**(1) Behavior Cloning.** We first perform behavior cloning (BC) on supervised interaction traces to initialize the policy within a reasonable behavioral manifold. Specifically, we construct a multi-turn SFT dataset by deploying the Qwen3 series model (Yang et al., 2025) in the target training environments, collecting self-generated interaction trajectories, and retaining only high-scoring rollouts for supervision. This self-bootstrapped SFT stage initializes the policy within a reasonable behavioral manifold aligned with the environment dynamics.

**(2) Format Penalty.** We incorporate $\mathcal{R}_{\text{format}}$ that enforces structured outputs with explicit `<think> </think>` and `<action> </action>` tags. If the generated output violates this format (*e.g.*, missing tags, malformed nesting, or extraneous content outside the tags), we apply a fixed penalty to the final reward. This explicit structural constraint provides dense shaping signals during early training and substantially reduces invalid rollouts that would otherwise corrupt policy updates.

**(3) Auxiliary KL Loss.** Unconstrained updates may cause the policy to drift excessively from the reference model. To regularize policy updates and preserve the pretrained knowledge embedded in the base model, we introduce a KL divergence penalty between the current policy $\pi_\theta$ and a reference policy $\pi_{\text{ref}}$. This constraint encourages conservative policy improvement while still allowing sufficient exploration in the action space. We adopt the commonly used Bregman divergence estimator $k_3$ for KL approximation, which leverages control variates to achieve unbiasedness and low variance (Schulman, 2017). Specifically, $k_3$ is defined as $k_3(x) = \delta(x) - 1 - \log \delta(x)$, where $\delta(x) = \frac{p(x)}{q(x)}$ denotes the likelihood ratio.

**(4) PO-specific Hyper-parameter Grid Search.** A natural question is how to ensure that each PO method is fairly evaluated in the multi-turn setting. Our solution is to first run each method with its default configuration, and then perform a PO-specific hyperparameter grid search. We continue tuning until the training trajectory becomes stable, measured by the variance of the success rate over the final 20% of training steps falling below a predefined threshold. As shown in Table 1, hyperparameters related to IS clipping are particularly sensitive. The best-performing configurations and full results are reported in Appendix B.

### 3.2. Tasks and Training Details

We adapt ALFWorld (Shridhar et al., 2020), Web-Shop (Yao et al., 2022), Sokoban (Schrader, 2018), and TIR Math (Xue et al., 2025) as the agentic tasks. Our entire codebase is built upon the `verl` RL framework (Sheng et al., 2025). We employ an agentic-loop architecture to coordinate rollouts and environment interactions, after which we segment each complete trajectory into multiple single-turn samples for policy optimization. For mathematical tasks, we use `Qwen3-4B-base` as the policy model, while for all other tasks we initialize from the SFT-tuned variant `Qwen3-4B`. For consistency validation, we additionally employed SFT-tuned `Qwen3-8B`, and the corresponding experimental results are provided in Appendix C. All experiments are conducted on NVIDIA H200 or B200 GPUs. Key hyperparameters and training details are reported in Appendix B.

## 4. Exploring Gradient Dimensions on ARL

The experimental results for all policy optimization methods are reported in Table 3. $GRPO_{ST}$ denotes GRPO with sequence-mean-token-mean loss aggregation. $DAPO_{GRPO}$ and $DAPO_{GIGPO}$ denote GRPO and GIGPO augmented with dynamic filtering, respectively.

### 4.1. Impact of IS on ARL

We study GSPO, CISPO, and SAPO along the importance-sampling (IS) dimension. GSPO adopts sequence-level clipping, while CISPO and SAPO employ tolerant clipping techniques. For CISPO and SAPO, we further apply sequence masking (denoted as $CISPO_{SM}$ and $SAPO_{SM}$) to improve training stability. Detailed training dynamics are reported in Figure 2, with IS token-level and sequence-level analyses presented in Figure 3. Table 3 shows that CISPO and SAPO perform substantially worse than GRPO across all tasks, achieving average scores of 34.03 and 32.22, respectively, compared to 46.16 for GRPO. In contrast, GSPO consistently outperforms all other policy optimization methods, achieving an average improvement of 13.3% compared to GRPO.

To understand training behavior beyond final performance, we analyze training dynamics from multiple perspectives across several metrics. Different IS designs induce varying distances between the current policy and both the behavior and reference policies during training. These distance variations, in turn, influence optimization behavior (reflected by gradient norms), impact data quality (through the valid action ratio), and ultimately affect task success rates. Jointly examining these metrics enables a more comprehensive understanding of training stability and failure modes. Figure 2 reports success rate, off-policy KL di-

vergence (between the new and old policies), KL loss (between the new and reference policies), gradient norm, and the valid-format ratio of rollout action tokens.

As shown in Figure 2, CISPO and SAPO with tolerant clipping exhibit rapid initial performance gains, characterized by higher success rates, larger policy updates relative to the reference model, and faster format ratio adaptation compared to GRPO and GSPO. This behavior indicates more aggressive optimization that departs quickly from the reference policy and adapts rapidly to the task. A possible explanation is that tolerant clipping may preserve gradient contributions from tokens that deviate substantially from the current policy, resulting in overly exploratory updates. However, such aggressiveness leads to training instability, with collapse occurring around step 130. This collapse is marked by exploding gradient norms and KL divergence, accompanied by a sharp drop in the valid-format ratio, ultimately resulting in a severe degradation of success rate. In contrast, GSPO demonstrates a substantially more stable training pattern, with gradual performance improvement accompanied by steady KL divergence and gradient norms. These results indicate that sequence-level clipping is effective for stabilizing training, while overly tolerant clipping thresholds may yield short-term gains at the cost of long-term stability. Furthermore, IS design substantially impacts both performance and training stability in ARL, making it an important dimension in ARL system design.

> **Finding 1**
>
> ARL is highly sensitive to IS design: tolerant clipping yields fast early gains but causes training collapse, whereas sequence-level clipping ensures stable improvement.

**Rooted cause of training collapse.** To investigate the root causes of training collapse along the IS dimension, we analyze token-level importance ratio statistics and stratify sequences by IS ratio, advantage, and entropy for SAPO and $SAPO_{SM}$, where $SAPO_{SM}$ denotes a stabilized variant of SAPO introduced later. Figure 3 reports token-level and sequence-level IS ratio analysis. Subfigures (a) and (b) present the statistics of tokens whose importance sampling ratios fall outside the standard clipping range. Specifically, we report the proportion of out-of-bounds tokens and decompose it into lower- and upper-bound portions. The lower-bound portion corresponds to negative-advantage tokens with importance ratios below $\epsilon_{low}$, while the upper-bound portion corresponds to positive-advantage tokens with ratios exceeding $\epsilon_{high}$.

As shown in Figure 3, during the collapse stage, SAPO exhibits a rapidly growing number of out-of-bounds tokens, predominantly from negative-advantage sequences with

| Dimension | Method | ALFWorld | | WebShop | | Sokoban | | TIR Math | | Avg |
| | | Score | Success | Score | Success | Score | Success | AIME | AIME25 | |
|---|---|---|---|---|---|---|---|---|---|---|
| Base | GRPO | 3.70 | 62.36 | 75.32 | 57.71 | 5.51 | 83.90 | 49.96 | 30.78 | 46.16 (48.08) |
| Loss Agg | GRPO$_{ST}$ | 4.41$_{\uparrow 19.2\%}$ | 72.61$_{\uparrow 16.4\%}$ | 64.57$_{\downarrow 14.3\%}$ | 51.29$_{\downarrow 11.1\%}$ | 3.03$_{\downarrow 45.0\%}$ | 68.73$_{\downarrow 18.1\%}$ | 27.55$_{\downarrow 44.9\%}$ | 21.63$_{\downarrow 29.7\%}$ | 39.23$_{\downarrow 15.0\%}$ |
| Importance Sampling | SAPO | 0.80$_{\downarrow 78.4\%}$ | 25.16$_{\downarrow 59.7\%}$ | 73.85$_{\downarrow 1.9\%}$ | 52.10$_{\downarrow 9.7\%}$ | −0.23$_{\downarrow 104\%}$ | 30.25$_{\downarrow 63.9\%}$ | 45.00$_{\downarrow 9.9\%}$ | 30.85$_{\uparrow 0.2\%}$ | 32.22$_{\downarrow 30.2\%}$ |
| | CISPO | 2.16$_{\downarrow 41.6\%}$ | 54.42$_{\downarrow 12.7\%}$ | 67.96$_{\downarrow 9.8\%}$ | 54.71$_{\downarrow 5.2\%}$ | −0.47$_{\downarrow 109\%}$ | 26.02$_{\downarrow 69.0\%}$ | 36.53$_{\downarrow 26.9\%}$ | 30.87$_{\uparrow 0.3\%}$ | 34.03$_{\downarrow 26.3\%}$ |
| | GSPO | 5.19$_{\uparrow 40.3\%}$ | 78.61$_{\uparrow 26.1\%}$ | 85.29$_{\uparrow 13.3\%}$ | 72.48$_{\uparrow 25.6\%}$ | 5.22$_{\downarrow 5.3\%}$ | 82.22$_{\downarrow 1.7\%}$ | 51.29$_{\uparrow 2.7\%}$ | 37.95$_{\uparrow 23.3\%}$ | 52.28$_{\uparrow 13.3\%}$ |
| Advantage Design | GIGPO | 4.97$_{\uparrow 34.3\%}$ | 81.09$_{\uparrow 30.0\%}$ | 67.76$_{\downarrow 10.0\%}$ | 56.55$_{\downarrow 2.0\%}$ | 5.19$_{\downarrow 5.8\%}$ | 82.67$_{\downarrow 1.5\%}$ | – | – | 49.71$_{\uparrow 3.4\%}$ |
| | EMPG | 3.32$_{\downarrow 10.3\%}$ | 57.91$_{\downarrow 7.1\%}$ | 79.16$_{\uparrow 5.1\%}$ | 64.32$_{\uparrow 11.5\%}$ | 4.48$_{\downarrow 18.7\%}$ | 79.16$_{\downarrow 5.6\%}$ | – | – | 48.06$_{\downarrow 0.1\%}$ |
| Dynamic Sampling | DAPO$_{GRPO}$ | 1.95$_{\downarrow 47.3\%}$ | 49.58$_{\downarrow 20.5\%}$ | 62.43$_{\downarrow 17.1\%}$ | 46.17$_{\downarrow 20.0\%}$ | 5.16$_{\downarrow 6.4\%}$ | 82.40$_{\downarrow 1.8\%}$ | 54.66$_{\uparrow 9.4\%}$ | 38.97$_{\uparrow 26.6\%}$ | 42.67$_{\downarrow 7.6\%}$ |
| | DAPO$_{GIGPO}$ | 2.49$_{\downarrow 32.7\%}$ | 60.55$_{\downarrow 2.9\%}$ | 88.10$_{\uparrow 17.0\%}$ | 76.82$_{\uparrow 33.1\%}$ | 6.01$_{\uparrow 9.1\%}$ | 86.20$_{\uparrow 2.7\%}$ | – | – | 53.36$_{\uparrow 11.0\%}$ |
| **Ours** | **SAMPO** | **7.04**$_{\uparrow 90.3\%}$ | **92.72**$_{\uparrow 48.7\%}$ | **88.37**$_{\uparrow 17.3\%}$ | **77.73**$_{\uparrow 34.7\%}$ | **6.56**$_{\uparrow 19.1\%}$ | **88.86**$_{\uparrow 5.6\%}$ | – | – | **60.21**$_{\uparrow 25.2\%}$ |

*Table 3.* Performance comparison of policy optimization methods across four agentic tasks, evaluated on the SFT version of Qwen3-4B. Methods are organized by their primary design dimension: loss aggregation, importance sampling clipping, advantage design, and dynamic sampling. Green/red subscripts denote the percentage improvement/degradation relative to the GRPO baseline. SAMPO (ours) achieves the highest average score (59.55) with consistent gains across ALFWorld (92.72% success), WebShop (74.08% success), and Sokoban (88.86% success). The evaluation metric for TIR Math is Pass@4; "–" indicates the method is not applicable. For GRPO, the value in parentheses reports the average over the first three tasks only.

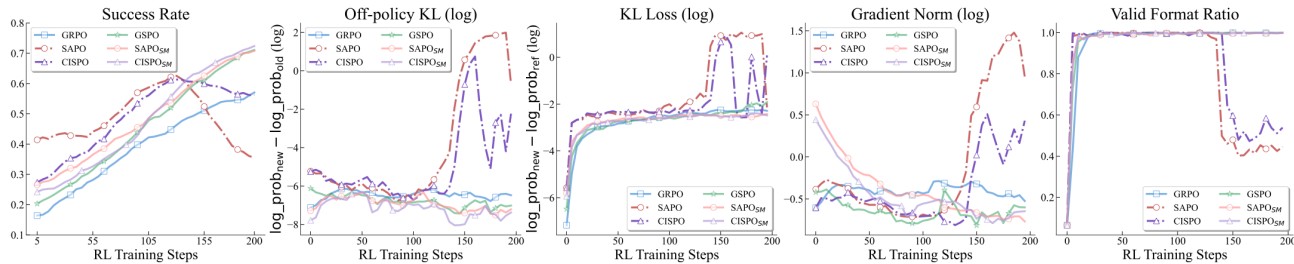

*Figure 2.* Training dynamics of six IS variants on ALFWorld: GRPO, GSPO, SAPO, CISPO, and their sequence-masked counterparts SAPO$_{SM}$ and CISPO$_{SM}$. Panels show (from left to right) success rate, off-policy KL divergence between the current and behavior policies, KL loss between the current and reference policies, gradient norm, and valid-format ratio of rollout actions.

small importance ratios (the lower-bound portion). In contrast, for stable training runs, the portion of out-of-bounds tokens remains fairly low, and lower- and upper-bound ratio portions remain relatively balanced. This growing pattern and imbalance during collapse suggests that negative-advantage samples with low IS ratios are the main contributors the observed training instability.

Beyond token-level analysis, we conduct a sequence-level comparison across training steps in Subfigures (c) and (d). Rollout samples are partitioned according to three factors: the sign of the advantage, whether the importance ratio is smaller or larger than one, and whether policy entropy falls below or exceeds a predefined threshold. This yields eight groups per training step. The vertical area denotes the normalized KL divergence between the current policy and the reference policy. A larger area therefore corresponds to a greater deviation from the reference policy, indicating a stronger contribution to policy shift during training. For collapsed experiments, the proportion of KL divergence attributed to sequences with negative advantages and low importance ratios increases abruptly, whereas for stable training this KL distribution remains relatively balanced across

groups. Entropy is less impactful than advantage and IS ratio. This pattern further reinforces the conclusion that negative-advantage samples with low importance ratios are a primary source of training instability.

**Stabilization Strategies for SAPO and CISPO.** We explore several strategies to stabilize SAPO and CISPO training, reported in Table 4. First, we consider increasing the KL coefficient to regularize optimization, and enlarging the mini-update batch size to mitigate off-policy effects. As shown in Table 4, increasing the KL coefficient overly constrains training and yields limited performance gains (full success-rate plots reported in Appendix C). Similarly, increasing the mini-update batch size degrades performance. Motivated by the IS-token analysis during training collapse, we adopt sequence masking following (Liu et al., 2025) to directly control negative samples that induce instability. Specifically, sequences with negative advantages and low importance ratios are masked (see Table 1 for the detailed formulation), a variant we denote as GRPO$_{SM}$. We apply sequence masking to SAPO and CISPO, denoted as SAPO$_{SM}$ and CISPO$_{SM}$. According to Figure 3 and Table 4, applying sequence masking improves the success rate from

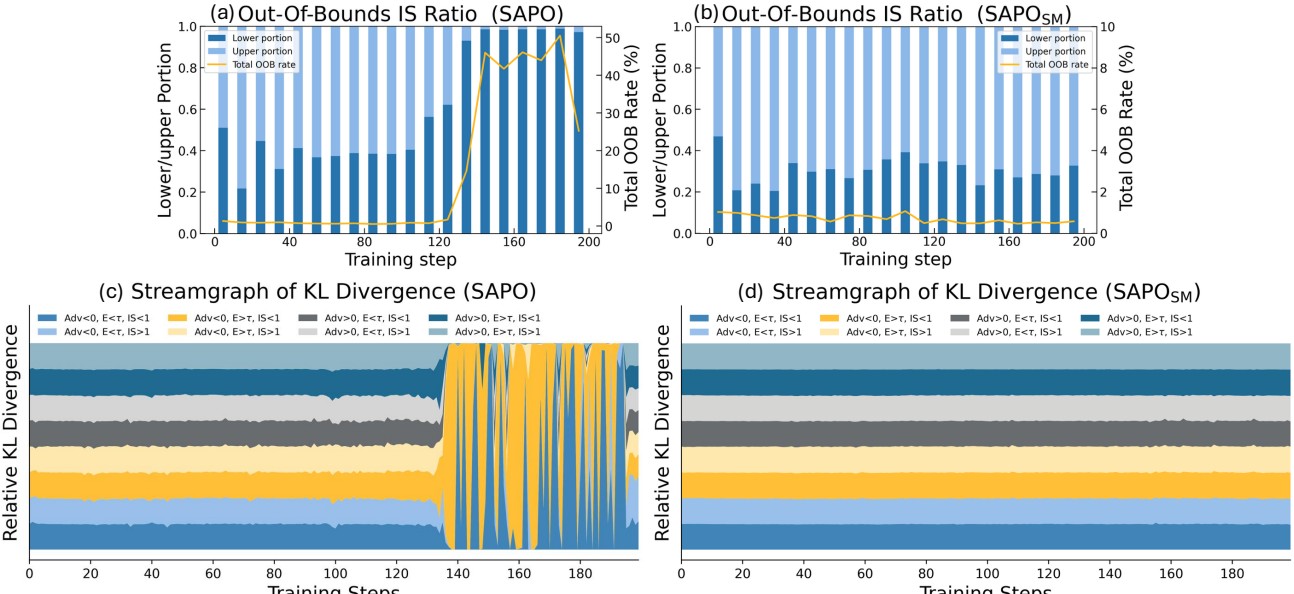

*Figure 3.* Token-level and sequence-level IS analysis of SAPO and its sequence-masked variant SAPO$_{SM}$. (a, b) Fraction of tokens with importance ratios outside the clipping range, decomposed into lower-bound (negative advantage) and upper-bound (positive advantage) portions. (c, d) Rollout groups partitioned by advantage sign, entropy level, and IS ratio magnitude, with KL divergence normalized for relative comparison.

| Method | Metric | Original | KL (0.05) | Off-Policy (1024) | Seq-Mask |
|--------|--------|----------|-----------|-------------------|----------|
| CISPO  | Score   | 2.16  | 1.60  | 0.98  | **5.25** |
|        | Success | 54.42 | 38.46 | 21.59 | **78.88** |
| SAPO   | Score   | 0.80  | 2.40  | 3.82  | **4.88** |
|        | Success | 25.16 | 48.05 | 64.30 | **76.92** |

*Table 4.* Effect of different stabilization strategies on CISPO and SAPO in ALFWorld. We evaluate three stabilization techniques applied to the tolerant-clipping methods CISPO and SAPO: increasing the KL penalty coefficient to 0.05, enlarging the off-policy mini-update batch size to 1024, and applying sequence-level masking (Seq-Mask).

54.12 to 78.88 for CISPO and from 25.16 to 76.92 for SAPO. SAPO$_{SM}$ and CISPO$_{SM}$ effectively stabilizes training, yielding success rates comparable to GSPO, along with steady KL divergence and gradient norms (Figures 2, 3).

> **Finding 2**
>
> Training collapse is largely driven by the accumulation of negative-advantage sequences with low IS ratios. Sequence masking of such sequences stabilizes training.

### 4.2. Impact of Advantage Design on ARL

We study GIGPO and EMPG along the advantage-design dimension. GIGPO incorporates both global and local advantage information from the environment, enabling fine-grained advantage estimation, while EMPG reshapes advantages by incorporating uncertainty information from the training data.

Table 3 shows that GIGPO generally outperforms GRPO, achieving an average score of 49.71 compared to 48.08,

with a particularly strong improvement of 34.4% on ALF-World. In addition, EMPG exhibits task-dependent performance, improving the success rate on WebShop by 11.5% while degrading performance on ALFWorld by 7.1%, resulting in an average score difference of 0.1 compared to GRPO. This suggests that fine-grained advantage design incorporating richer environmental information improves performance and alleviates reward sparsity in ARL, whereas advantage reshaping based on uncertainty signals has a smaller effect.

> **Finding 3**
>
> Incorporating fine-grained environmental advantage in ARL improves performance.

### 4.3. Impact of Dynamic Filtering on ARL

Dynamic filtering is well known for delivering strong performance improvements on mathematical reasoning tasks (Yu et al., 2025a; Xue et al., 2025). However, we find that these gains do not always transfer to ARL settings. As shown in Table 3, dynamic filtering improves performance more consistently when combined with GIGPO than with GRPO. This difference stems from how dynamic filtering interacts with format learning. In early training, many rollout groups fail entirely due to format errors, which amplifies the format penalty and produces strong implicit advantage signals for format correction. As a result, the model rapidly acquires correct formatting from early rollouts. Meanwhile, dynamic filtering removes such all-failure groups. For GRPO, whose advan-

| Degree | ALFWorld | | Math | | | |
| | Score | Success | AIME | | AIME25 | |
| | | | k@1 | k@32 | k@1 | k@32 |
|---|---|---|---|---|---|---|
| Low | 3.50 | 60.80 | 26.95 | 87.34 | 24.61 | 50.00 |
| Medium | 3.83 | 58.38 | 24.22 | 75.00 | 17.97 | 48.59 |
| High | 2.33 | 52.71 | 19.53 | 74.99 | 16.41 | 43.85 |

*Table 5.* Effect of off-policy staleness on ALFWorld and MATH. We vary the degree of off-policy staleness (Low, Medium, High) and report task score, success rate (ALFWorld), and pass@$k$ accuracy (AIME, AIME25).

tage signals have limited diversity, filtering substantially reduces format-related learning signals, leading to unstable format behavior and limited gains. In contrast, GIGPO produces more diverse advantage signals, which stabilize format learning even after filtering, allowing DAPO$_{\text{GIGPO}}$ to achieve better and more stable performance. The detailed evidence is provided in Appendix G.

> **Finding 4**
>
> Dynamic filtering with GIGPO is beneficial for training stability and performance in ARL.

### 4.4. Impact of Loss Aggregation on ARL

As shown in Table 3, sequence-mean-token-mean loss aggregation (GRPO$_{\text{ST}}$) degrades performance from 46.16 to 39.23 relative to token-mean aggregation (GRPO). Although GRPO$_{\text{ST}}$ yields a 16.4% improvement on ALFWorld, it leads to a substantial decline on TIR-Math, with a 44.9% decrease on AIME. Notably, math rollouts exhibit higher variance in sequence length compared to other tasks, ranging from brief solutions to extended reasoning traces. These findings suggest that the unbalanced token weighting induced by sequence-level aggregation may negatively affect ARL training, particularly in tasks characterized by high length variability.

### 4.5. Further Stability Considerations

**Exploration on Off-Policy Staleness.** Due to infrastructure and efficiency constraints, policy training is typically performed in batched rollouts, where groups of trajectories are generated and updated sequentially before proceeding to the next rollout stage. Off-policy effects arise because later updates within the same rollout stage use data from an earlier policy while the current policy has already evolved. Such off-policy mismatch is further amplified in multi-turn settings, where turn-wise decomposition increases the number of samples subject to staleness.

**Experiment Setup and Results.** We control off-policy degree through rollout configuration while holding the update batch size fixed. For TIR Math, rollout batch sizes of

128, 512, and 1024 correspond to low, medium, and high off-policy degrees, respectively. For ALFWorld, we vary the off-policy degree by adjusting the number of groups per rollout to 8, 16, and 32. The effects of off-policy staleness are summarized in Table 5. TIR Math achieves higher performance under a low off-policy ratio (rollout batch size = 128), with 87.34% and 50.00% for pass@32, compared to 74.99% and 43.85% under a high off-policy ratio. Similarly, ALFWorld attains its highest success rate of 60.80% under low off-policy settings, which decreases to 52.71% under high off-policy settings. These results suggest that policy gradient optimization for agentic tasks exhibits sensitivity to the off-policy ratio.

## 5. SAMPO

### 5.1. Motivation

*Can we derive a unified understanding of ARL training based on these insights?* By systematically analyzing POs along orthogonal design dimensions in ARL, we identify key factors that determine training stability and optimization efficacy. At initialization, formatting errors and invalid action tokens induce severe optimization noise. We eliminate these failure modes through behavior cloning and explicit format correction, constraining learning to a valid behavioral manifold. Along the importance sampling dimension, sequence-level clipping, rather than token-wise constraints, is critical for long-horizon ARL. This mechanism addresses off-policy drift by suppressing harmful trajectories and yields substantial improvements in training stability. For advantage design, our analysis reveals that increasing advantage diversity across finer scales is essential to overcoming reward sparsity. Integrating global and local signals significantly enhances credit assignment. Finally, we show that dynamic trajectory filtering helps stabilize gradient updates by removing samples with degenerate advantages, leading to more informative policy gradients.

### 5.2. Our Method

Guided by this unified understanding, we propose SAMPO, a new PO paradigm built on these principles. SAMPO integrates sequence-level clipping, fine-grained advantage estimation, and dynamic filtering into a unified framework, yielding a stable and scalable solution for ARL. It is formulated as:

$$\mathcal{L}(\theta) = \frac{1}{\sum_{i=1}^{N} T_i} \sum_{i=1}^{N} \sum_{t=0}^{T_i-1} \min\left(s_i(\theta) A_i^{'},\ \text{clip}\left(s_i(\theta), 1\pm\varepsilon\right) A_i^{'}\right),$$
$$\text{s.t.} \quad 0 < |\{y \mid \text{is\_equivalent}(a, y)\}| < G.$$

$$(8)$$

Here, $A_{i,k}^{'} = A_i + \omega \cdot A_{\text{step}}(\hat{y}_{i,k})$, $s_i(\theta) = \exp\left(\frac{1}{|T_i|} \sum_{t=0}^{|T_i|-1} \log \frac{\pi_\theta(y_t|x,y_{<t})}{\pi_{\theta_{\text{old}}}(y_t|x,y_{<t})}\right)$. Across all evaluated agentic tasks, SAMPO consistently achieves the

strongest overall performance shown in Table 1. Compared to methods that modify only one dimension, SAMPO demonstrates that combining multiple design dimensions is necessary for stable and effective ARL. Notably, SAMPO delivers particularly large improvements on long-horizon interactive tasks such as ALFWorld, highlighting the importance of sequence-aware control in agentic settings. These results validate our central claim that stable agentic PO method requires satisfying multiple necessary conditions simultaneously, rather than relying on isolated algorithmic modifications.

### 5.3. Benchmarking against Inference Paradigms

To further contextualize the performance of SAMPO and evaluate whether a small open-source model trained with stable RL can compete with state-of-the-art inference strategies, we benchmark ARLArena against frontier closed-source models and complex multi-agent workflows. This comparison verifies a key hypothesis: principled RL training may offer greater gains in agentic tasks than heavy inference-time engineering on generic models.

**Experiment Setup and Results.** We evaluate GPT-5.2 (OpenAI, 2025a), o3 (OpenAI, 2025b), and Gemini 2.5 Pro (Comanici et al., 2025) on ALFWorld and WebShop, under two paradigms: (i) Single LLM as Agent (SLA), following a standardized protocol; (ii) Multi-Agent System (MAS), with Debate and Aggressive Debate coordination strategies (details are revealed in Appendix F.5). Qwen3-4B-RFT post-trained with SAMPO achieves 92.72% all-task success on ALFWorld, outperforming GPT-5.2 (51.56%) and o3-based MAS (56.25%). Open-source models with SAMPO consistently exceed larger closed-source models, showing that scale and complex inference cannot replace stable, environment-aligned ARL training.

## 6. Insights for Future Work

Based on our systematic dissection of policy gradient design choices in ARL, we identify several promising directions that merit deeper exploration.

**(1) Clean training recipes are foundational for complex reasoning.** ARLArena reveals that ARL is extraordinarily sensitive to initialization and early-stage training dynamics. A carefully constructed clean setting, combining short supervised cold-start SFT, format-enforcing structural constraints, and conservative KL regularization, proves essential for unlocking stable multi-turn reasoning behaviors. Without such a controlled recipe, policy gradient signals are easily corrupted by malformed trajectories or premature collapse. This suggests that future research should treat training recipes not as auxiliary tricks, but as essential algorithmic components that define the

feasible region in which sophisticated reasoning policies can emerge. Our codebase also provides detailed training recipes for reference.

**(2) IS clipping is highly sensitive, while advantage design offers a comparatively stable gain.** Among the policy gradient dimensions we examine, IS clipping strategies exhibit high sensitivity: minor changes in clipping thresholds or ratio parameterization can drastically affect stability. In contrast, advantage design tends to provide more stable but relatively modest improvements across tasks. These observations indicate that IS clipping strategy represents a *high-risk, high-reward* direction, whereas advantage design offers a more predictable but limited performance gains in ARL.

**(3) Stable ARL unlocks long-horizon scaling opportunities.** Once training collapse is mitigated, we observe that agentic policies can sustain performance improvements over substantially more optimization steps without degradation. This stability opens the door to scaling both interaction horizon and environment size, analogous to scaling laws in supervised pretraining. Consequently, future progress in the field will increasingly depend on scaling environment diversity, interaction data volume, and multi-task curricula.

## 7. Conclusion

This work systematically analyzes how policy gradient design choices impact training stability for agentic LLMs in multi-turn environments. ARLArena demonstrates that sequence-level clipping is critical for stability, while advantage design and dynamic filtering offer smaller but consistent gains, and loss aggregation has limited effect. Based on these insights, we introduce **SAMPO**, a unified policy optimization framework that achieves stable and effective agentic RL training. Overall, this study underscores the importance of principled policy design and reproducible evaluation for advancing ARL.

## Acknowledgements

This work was partially supported by the National Science Foundation (NSF 2303037, 2312501, 2531008, 2106859), the National Institutes of Health (OT2OD038003, U54OD036472, U54HG012517, U54DK097771), Defense Advanced Research Projects Agency (DARPA HR001126CE054), SRC JUMP 2.0 Center, Amazon, Amazon Research Awards, NEC, Optum AI, Snapchat, Google Gifts, AMD AI and HPC Cluster. We also acknowledge the Center for High Throughput Computing (CHTC) at the University of Wisconsin–Madison for providing computational resources and support for

supplementary analyses in the appendix (Center for High Throughput Computing, 2006).

## Impact Statement

This work aims to improve the stability, reproducibility, and transparency of ARL for LLM–based agents. By providing a unified policy-gradient perspective and a systematic analysis framework, our study facilitates more reliable development and evaluation of agentic learning algorithms, which can benefit a wide range of applications involving multi-step decision making, such as automated information seeking, planning, and interactive problem solving.

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

# Supplementary Materials for ARLArena

## A. More Details on Research Dimension

### A.1. Loss Aggregation

As discussed in Section 2.2, the policy gradient objective for agentic LLMs is implemented through a batch-level loss aggregation over token-level surrogate losses. For a batch of $N$ sampled trajectories $\{y_i\}_{i=1}^N$, where trajectory $i$ has length $T_i$, we define the token-level loss as

$$\ell_{i,t}(\theta) := \min\left(w_{i,t}(\theta)A_i,\ \mathrm{clip}(w_{i,t}(\theta), 1 - \varepsilon, 1 + \varepsilon)\, A_i\right), \tag{9}$$

where $w_{i,t}(\theta) = \pi_\theta(y_{i,t} \mid x_i, y_{i,<t})/\pi_{\theta_{\mathrm{old}}}(y_{i,t} \mid x_i, y_{i,<t})$ and $A_i$ denotes the (sequence-level) advantage associated with trajectory $y_i$.

Different loss aggregation strategies correspond to different empirical estimators of the expectation over trajectories and tokens. Below we summarize several commonly used schemes.

**Token-mean.** The token-mean estimator averages the loss uniformly over all unmasked tokens in the batch:

$$\mathcal{L}_{\text{token-mean}}(\theta) = \frac{1}{\sum_{i=1}^N T_i} \sum_{i=1}^N \sum_{t=0}^{T_i-1} \ell_{i,t}(\theta). \tag{10}$$

This scheme assigns equal weight to each token across the entire batch and is invariant to trajectory length at the sequence level. Token-mean has been adopted in several recent works (e.g., DAPO) as a means of stabilizing optimization. However, because trajectories with longer responses contribute more tokens, they implicitly receive larger total weight, which may bias optimization toward long trajectories.

**Sequence-mean token-mean (Seq-mean-token-mean).** This estimator first averages over tokens within each trajectory and then averages across trajectories:

$$\mathcal{L}_{\text{seq-mean-token-mean}}(\theta) = \frac{1}{N} \sum_{i=1}^N \frac{1}{T_i} \sum_{t=0}^{T_i-1} \ell_{i,t}(\theta). \tag{11}$$

Under this scheme, each trajectory contributes equally regardless of its length. Equivalently, each token is weighted by $1/T_i$. As a result, shorter trajectories assign larger per-token weight, while longer trajectories are relatively down-weighted. This behavior can introduce response-level length bias, rewarding short correct trajectories more strongly and penalizing long incorrect trajectories less.

**Sequence-mean token-sum (Seq-mean-token-sum).** An alternative aggregation removes the per-trajectory normalization over tokens:

$$\mathcal{L}_{\text{seq-mean-token-sum}}(\theta) = \frac{1}{N} \sum_{i=1}^N \sum_{t=0}^{T_i-1} \ell_{i,t}(\theta). \tag{12}$$

This formulation corresponds to maximizing the expected cumulative surrogate objective over full trajectories. Compared to Seq-mean-token-mean, longer trajectories receive proportionally larger weight.

**Sequence-mean token-sum with length normalization (Seq-mean-token-sum-norm).** In practice, some implementations normalize by a fixed maximum generation length $T_{\max}$:

$$\mathcal{L}_{\text{seq-mean-token-sum-norm}}(\theta) = \frac{1}{NT_{\max}} \sum_{i=1}^N \sum_{t=0}^{T_i-1} \ell_{i,t}(\theta). \tag{13}$$

This estimator enforces a uniform upper bound on the contribution of each trajectory and assigns equal weight to tokens across batches under a fixed-length budget.

**Discussion.** These aggregation schemes differ primarily in how they trade off trajectory-level fairness, token-level weighting, and variance control. Seq-mean-token-mean and token-mean are the two most commonly used estimators in practice and are the focus of our empirical analysis in Section 4.4. The remaining variants are included here for completeness and to clarify their implicit inductive biases in agentic reinforcement learning.

### A.2. Importance Sampling Clipping

As discussed in Section 2.2, importance sampling (IS) clipping plays a central role in stabilizing off-policy policy optimization. While all methods considered in this work rely on the same token-level importance ratio

$$w_{i,t}(\theta) = \frac{\pi_\theta(y_{i,t} \mid x_i, y_{i,<t})}{\pi_{\theta_{\mathrm{old}}}(y_{i,t} \mid x_i, y_{i,<t})}, \tag{14}$$

they differ substantially in *where* and *how* clipping is applied. Below we summarize the clipping mechanisms of GRPO, CISPO, SAPO, and GSPO.

#### A.2.1. GRPO

Group Relative Policy Optimization (GRPO) adopts the standard PPO-style hard clipping applied independently at each token:

$$\ell_{i,t}^{\text{GRPO}}(\theta) = \min\left(w_{i,t}(\theta)A_i,\ \mathrm{clip}(w_{i,t}(\theta), 1 - \varepsilon, 1 + \varepsilon)\, A_i\right). \tag{15}$$

Clipping is performed directly on the token-level importance ratio. When $w_{i,t}$ falls outside the clipping range, the gradient contribution of that token is truncated.

### A.2.2. CISPO

Clipped Importance Sampling Policy Optimization (CISPO) modifies GRPO by clipping the importance ratio itself rather than the surrogate objective. Specifically, the clipped ratio is defined as

$$
\tilde{w}_{i,t}(\theta) = \begin{cases} 1 + \varepsilon, & w_{i,t}(\theta) > 1 + \varepsilon, \\ w_{i,t}(\theta), & \text{otherwise,} \end{cases} \tag{16}
$$

and is treated as a stop-gradient quantity. The resulting loss takes the form

$$
\ell_{i,t}^{\text{CISPO}}(\theta) = \text{sg}(\tilde{w}_{i,t}(\theta)) \, A_i \, \log \pi_\theta(y_{i,t} \mid x_i, y_{i,<t}), \tag{17}
$$

where $\text{sg}(\cdot)$ denotes the stop-gradient operator. By avoiding hard truncation of token updates, CISPO preserves gradient flow for clipped tokens while still bounding their influence. However, clipping remains token-local and does not explicitly enforce sequence-level coherence.

### A.2.3. SAPO

Soft Adaptive Policy Optimization (SAPO) replaces hard clipping with a smooth, temperature-controlled gating function. The surrogate loss is defined as

$$
\ell_{i,t}^{\text{SAPO}}(\theta) = f_{i,t}(w_{i,t}(\theta)) \, A_i, \tag{18}
$$

where

$$
f_{i,t}(x) = \sigma(\tau_{i,t}(x-1)) \cdot \frac{4}{\tau_{i,t}}, \qquad \tau_{i,t} = \begin{cases} \tau_{\text{pos}}, & A_i > 0, \\ \tau_{\text{neg}}, & A_i < 0. \end{cases} \tag{19}
$$

Here $\sigma(\cdot)$ denotes the sigmoid function. SAPO implements a continuous trust region: near on-policy updates are preserved, while off-policy updates are smoothly attenuated rather than abruptly clipped. The asymmetric temperature design further suppresses high-variance negative-advantage updates. Despite improved smoothness, SAPO remains a token-level method and does not explicitly prevent a few extreme tokens from destabilizing a full trajectory.

### A.2.4. GSPO

Group Sequence Policy Optimization (GSPO) fundamentally changes the unit of clipping by operating at the sequence level. The sequence-level importance ratio is defined as

$$
s_i(\theta) = \exp\left(\frac{1}{T_i} \sum_{t=0}^{T_i-1} \log w_{i,t}(\theta)\right) = \left(\frac{\pi_\theta(y_i \mid x_i)}{\pi_{\theta_{\text{old}}}(y_i \mid x_i)}\right)^{1/T_i}. \tag{20}
$$

Clipping is then applied once per sequence:

$$
\ell_i^{\text{GSPO}}(\theta) = \min(s_i(\theta)A_i, \, \text{clip}(s_i(\theta), 1 - \varepsilon, 1 + \varepsilon) \, A_i). \tag{21}
$$

All tokens within a trajectory share the same clipped update. This design aligns the unit of importance sampling with the unit of reward and enforces strong sequence-level coherence. As a result, GSPO effectively suppresses high-variance token outliers and yields substantially more stable optimization in long-horizon agentic reinforcement learning.

**Summary.** In summary, GRPO, CISPO, and SAPO apply clipping at the token level with increasing degrees of smoothness, whereas GSPO performs clipping at the sequence level. Our empirical results in Section 4.1 demonstrate that sequence-level clipping is a key factor for stabilizing multi-turn agentic RL training.

### A.3. Advantage Design

This section provides detailed formulations of the advantage designs introduced in Section 2.2, including Group-in-Group Policy Optimization (GiGPO) and Entropy-Modulated Policy Gradients (EMPG). Both methods extend standard group-based advantage estimation to better handle long-horizon agentic reinforcement learning.

**Notation.** We consider a batch of $N$ trajectories $\{\tau_i\}_{i=1}^N$, where each trajectory $\tau_i = \{(s_{i,k}, a_{i,k}, r_{i,k})\}_{k=1}^{K_i}$ is generated under the behavior policy $\pi_{\theta_{\text{old}}}$. The total return of a trajectory is denoted by

$$
R(\tau_i) = \sum_{t=1}^{T_i} r_{i,k}. \tag{22}
$$

### A.3.1. GROUP-IN-GROUP POLICY OPTIMIZATION (GiGPO)

GiGPO introduces a hierarchical advantage structure that combines trajectory-level and step-level relative advantages. The design preserves the critic-free and group-based nature of GRPO while enabling finer-grained credit assignment.

**Episode-level relative advantage.** GiGPO first computes a trajectory-level (episode-level) relative advantage by normalizing total returns within the rollout group:

$$
A_i = \frac{R(\tau_i) - \text{mean}\left(\{R(\tau_j)\}_{j=1}^N\right)}{F_{\text{norm}}\left(\{R(\tau_j)\}_{j=1}^N\right)}, \tag{23}
$$

where $F_{\text{norm}}(\cdot)$ is a normalization factor. In the original formulation, $F_{\text{norm}}$ may be chosen as the standard deviation or a fixed constant.

**Step-level relative advantage via anchor state grouping.** To assign fine-grained credit within a trajectory, GiGPO constructs step-level groups based on repeated environment states. Let $\mathcal{U}$ denote the set of distinct environment states appearing in the trajectory batch. For each anchor state $\tilde{s} \in \mathcal{U}$, a step-level group is defined as

$$\mathcal{G}_S(\tilde{s}) = \left\{ \left(a_{i,k}, R_{i,k}\right) \mid s_{i,k} = \tilde{s} \right\}, \qquad (24)$$

where $R_{i,k}$ denotes the discounted return from step $k$ reward:

$$R_{i,k} = \sum_{m=t}^{T_i} \gamma^{m-t} r_{i,m}. \qquad (25)$$

Within each step-level group, GiGPO computes a relative advantage for individual actions:

$$A_{\text{step}}(\hat{y}_{i,k}) = \frac{R_{i,k} - \text{mean}\left(\{R_{j,k'} \mid (a_{j,k'}, R_{j,k'}) \in \mathcal{G}_S(\tilde{s})\}\right)}{F_{\text{norm}}\left(\{R_{j,k'} \mid (a_{j,k'}, R_{j,k'}) \in \mathcal{G}_S(\tilde{s})\}\right)}. \qquad (26)$$

**Combined advantage.** The final advantage used for policy optimization is a linear combination of episode-level and step-level components:

$$A'_{i,k} = A_i + \omega \, A_{\text{step}}(y_{i,k}), \qquad (27)$$

where $\omega \geq 0$ is a weighting coefficient controlling the contribution of step-level credit.

### A.3.2. ENTROPY-MODULATED POLICY GRADIENTS (EMPG)

Entropy-Modulated Policy Gradients (EMPG) augments the advantage function by incorporating step-wise uncertainty measured via policy entropy. The method reshapes the learning signal at each decision step while preserving a trajectory-level optimization objective, making it suitable for long-horizon agentic reinforcement learning.

**Step-level entropy.** For a trajectory $\tau_i$ and its $t$-th step, EMPG defines a step-level entropy $H_{i,t}$ as the average token-level entropy over the tokens generated at that step:

$$H_{i,t} = -\frac{1}{|y_{i,t}|} \sum_{j=1}^{|y_{i,t}|} \sum_{v \in \mathcal{V}} \pi_\theta(v \mid y_{i,t,<j}) \log \pi_\theta(v \mid y_{i,t,<j}), \qquad (28)$$

where $|y_{i,t}|$ is the number of tokens in step $t$, $y_{i,t,<j}$ denotes the prefix before token $j$ within that step, and $\mathcal{V}$ is the vocabulary.

**Entropy-modulated advantage.** Let $A(\tau_i)$ denote the trajectory-level advantage (e.g., computed via group-based normalization as described in Section 2.2). EMPG defines a step-wise modulated advantage as

$$A_{\text{mod}}(i,t) = g(H_{i,t}) \, A(\tau_i) + \zeta \, f(H_{i,t+1}), \qquad (29)$$

where $g(\cdot)$ is a self-calibrating scaling function based on current-step entropy, $f(\cdot)$ is a future-clarity bonus depending on the next step, and $\zeta \geq 0$ controls the contribution of the future-clarity term.

**Self-calibrating gradient scaling.** The scaling function $g(\cdot)$ reweights the trajectory-level advantage according to the relative entropy of each step within a batch:

$$g(H_{i,t}) = \frac{\exp\left(-k\,\tilde{H}_{i,t}\right)}{\frac{1}{\sum_j T_j} \sum_{j,t'} \exp\left(-k\,\tilde{H}_{j,t'}\right)}, \qquad (30)$$

where $\tilde{H}_{i,t}$ denotes a batch-normalized entropy value, $T_j$ is the length of trajectory $\tau_j$, and $k > 0$ is a temperature parameter. This normalization ensures that the average scaling factor over the batch equals one.

**Future clarity bonus.** To encourage transitions toward lower-uncertainty future states, EMPG introduces a future-clarity bonus defined as

$$f(H_{i,t+1}) = \exp\left(-k'\,\tilde{H}_{i,t+1}\right), \qquad (31)$$

where $k' > 0$ controls sensitivity to the entropy of the next step.

**Final advantage normalization.** After computing $A_{\text{mod}}(i,t)$ for all steps in the batch, EMPG applies a final batch-level normalization (e.g., zero-mean normalization) before using the resulting advantages in policy gradient updates.

## B. Key Hyper-parameter

The hyperparameters reported in Table 6 are determined through task-specific grid search. For each policy optimization method and environment, we sweep over the method-relevant hyperparameters while keeping the remaining training and optimization settings fixed. The final configurations correspond to the stable settings selected from the grid search.

## C. Additional Experiment Result

### C.1. Performance on 8B Model

To further investigate the scalability of our findings, we evaluate the 8B parameter model (Qwen3-8B) on ALF-World, which serves as a representative benchmark for complex, multi-turn agentic tasks. Given the substantial computational requirements for large-scale RL training, we focus on this environment to verify if the core design principles distilled from the 4B models remain consistent at a larger scale.

*Table 6.* Key training hyperparameters for agentic RL experiments across four tasks (ALFWorld, WebShop, Sokoban, TIR Math). "–" indicates the method is not applicable to that task.

| Category | | ALFWorld | WebShop | Sokoban | TIR Math |
|---|---|---|---|---|---|
| *Model and Environment Configuration* | | | | | |
| Base model | | Qwen3-4B-RFT | Qwen3-4B-RFT | Qwen3-4B-VL-Instruct-RFT | Qwen3-4B-Base |
| Max interaction steps | | 50 | 15 | 15 | 5 |
| Memory context window | | 2 (turns) | 2 (turns) | 2 (turns) | 8196 (tokens) |
| Group rollout size | | 8 | 8 | 8 | 5 |
| Max prompt length | | 2048 | 4096 | 1024 | 8196 |
| Max response length | | 512 | 512 | 512 | 4096 |
| Format penalty coefficient | | 0.1 | 0.1 | 0.1 | 0.1 |
| *Training Optimization* | | | | | |
| Group normalization mode | | mean_std_norm | mean_std_norm | mean_std_norm | mean_std_norm |
| Learning rate | | $1 \times 10^{-6}$ | $1 \times 10^{-6}$ | $1 \times 10^{-6}$ | $1 \times 10^{-6}$ |
| Mini-batch size | | 256 | 128 | 64 | 128 |
| KL coefficient | | 0.01 | 0.01 | 0.01 | 0 |
| *Rollout and Inference Configuration* | | | | | |
| Rollout engine | | vLLM | vLLM | vLLM | vLLM |
| Temperature (training) | | 1.0 | 1.0 | 1.0 | 1.0 |
| Temperature (validation) | | 0.6 | 0.6 | 0.7 | 0.6 |
| Top-$p$ (validation) | | 0.95 | 0.95 | 0.95 | 0.95 |
| Top-$k$ (validation) | | 20 | 20 | 20 | 20 |
| *Training and Batching* | | | | | |
| User Prompt Number | | 16 | 16 | 32 | 512 |
| Validation batch size | | 128 | 128 | 128 | 128 |
| Total epochs | | 200($\sim$ 24h) | 200($\sim$ 22h) | 200($\sim$ 12h) | 17($\sim$ 60h) |
| GPUs | | NVIDIA H200/B200 | NVIDIA H200/B200 | NVIDIA H200/B200 | NVIDIA H200/B200 |
| *PO-specific Parameters* | | | | | |
| GRPO | $\varepsilon_{high}$ | 0.2 | 0.2 | 0.2 | 0.28 |
| | $\varepsilon_{low}$ | 0.2 | 0.2 | 0.2 | 0.2 |
| GIGPO | $\varepsilon$ | 0.2 | 0.2 | 0.2 | – |
| | $\gamma$ | 0.95 | 0.95 | 0.95 | – |
| | $\omega$ | 1 | 1 | 1 | – |
| EMPG | $\varepsilon$ | 0.2 | 0.2 | 0.2 | – |
| | $k, k'$ | 1.0 | 1.0 | 1.0 | – |
| | $\zeta$ | 0.05 | 0.05 | 0.05 | – |
| GSPO | $\varepsilon_{high}$ | 4e-3 | 4e-2 | 4e-3 | 4e-4 |
| | $\varepsilon_{low}$ | 3e-3 | 3e-2 | 3e-3 | 3e-4 |
| CISPO | $\varepsilon_{high}$ | 0.2 | 0.2 | 0.2 | 0.28 |
| | $\varepsilon_{low}$ | 1 | 1 | 1 | 1 |
| SAPO | $\tau_{pos}$ | 1.0 | 1.0 | 1.0 | 1.0 |
| | $\tau_{neg}$ | 1.05 | 1.05 | 1.05 | 1.05 |
| DAPO | $\varepsilon_{high}$ | 0.2 | 0.2 | 0.2 | 0.28 |
| | $\varepsilon_{low}$ | 0.2 | 0.2 | 0.2 | 0.2 |
| | $N_{oversample}$ | 3 | 3 | 3 | 3 |

As shown in Table 7, the experimental results on ALF-World and WebShop demonstrate that the relative performance gains and stability trends are highly consistent with our observations in the 4B experiments in Section 4. Specifically, the critical importance of sequence-level clipping is reaffirmed: even with increased model capacity, it remains the indispensable factor for preventing training collapse. Furthermore, we observe that the benefits of advantage design and dynamic filtering persist at this larger scale, providing consistent but incremental improvements to final performance. In contrast, the choice of loss aggregation continues to exhibit limited impact, echoing our findings on 4B models. These results collectively suggest that the hierarchical impact of policy design choices—and the resulting SAMPO recipe—is robust and scale-invariant, effectively leveraging the enhanced reasoning capabilities of larger models while maintaining stable training dynamics.

### C.2. Additional Analysis Result

We further visualize the training dynamics of CISPO and $\text{CISPO}_{\text{SM}}$ on the AlfWorld task using diagrams. Specifically, following the same setup as in the main text, we categorize trajectories according to three factors: the sign of the advantage, whether the entropy exceeds a predefined threshold, and whether the IS ratio is greater than zero. These criteria partition the samples into eight groups, which we use to analyze how the KL divergence evolves during training.

Consistent with our earlier findings, we clearly observe that after CISPO collapses, trajectories with negative advantages and low IS ratios (i.e., adv $< 0$ and IS $< 1$) rapidly dominate the distribution. This imbalance correlates strongly with the surge in KL divergence and subsequent training instability.

This observation also explains why $\text{CISPO}_{\text{SM}}$, which incorporates sequence-level masking, achieves substantially improved stability: by masking these harmful negative-advantage and low-ratio trajectories, the optimization process avoids pathological updates and maintains more balanced gradient signals.

### C.3. Task Environment Details

**ALFWorld (Shridhar et al., 2020)**: It provides a text-based interactive setting in which LLM agents are required to complete goal-driven tasks that involve reasoning over multiple sequential decisions. The environment focuses on everyday household activities and evaluates an agent's ability to plan and act through iterative interaction.

**WebShop (Yao et al., 2022)**: It is a large-scale interactive environment that places agents in realistic e-commerce scenarios, requiring them to interpret user instructions and make sequential decisions to identify and purchase suitable products.

**Sokoban (Schrader, 2018)**: It is a classic grid-based planning task where an agent navigates a 2D environment to push all boxes onto designated target cells. The state is represented visually, and the agent selects from discrete movement actions.

**TIR Math (Xue et al., 2025)**: This task focuses on standard mathematical question answering, where Python is used as a tool for intermediate calculations and symbolic reasoning. The overall pipeline follows Xue et al. (2025). The training data are adapted from SimpleRL (Zeng et al., 2025), and evaluation is conducted on the AIME and AIME25 benchmarks. Performance is measured using $\text{pass@k}$, following the evaluation protocol in Yu et al. (2025b).

## D. Related Work

Large language models have demonstrated strong capabilities in agent-based environments and attracted increasing attention (Yao et al., 2022; Shridhar et al., 2020; Li et al., 2023). Prior studies investigate LLMs as agents in multi-turn, action-based environments, emphasizing long-horizon memory and explicit tool use for sequential decision making and reasoning (Yao et al., 2023; Schick et al., 2023; Wang et al., 2023). Recently, driven by the success of reinforcement learning in reasoning (Xu et al., 2025; OpenAI, 2025a; Khatri et al., 2025), RL has been extended to agentic settings (Jin et al., 2025; Plaat et al., 2025; Abdulhai et al., 2023; Yu et al., 2025c). Several representative RL frameworks for LLM agents have emerged. AG-ILE (Peiyuan et al., 2024) proposes a framework for LLM-driven conversational agents capable of planning, tool use, and expert consultation. SWEET-RL (Zhou et al., 2025) studies collaborative LLM agents that interact with simulated human partners in ColBench, where agents ask clarifying questions and learn from multi-turn feedback. Agent-R1 (Cheng et al., 2025) extends this paradigm to external tool-based environments and enables multi-turn reasoning with tool calls. Similarly, AgentGym-RL (Xi et al., 2025) presents an RL framework for autonomous LLM agents that supports multi-turn interactions, modular architectures, and real-world scenarios. AgentRL (Zhang et al., 2025) develops a multi-turn, multi-task RL system and demonstrates superior performance relative to closed-source models. VerlTool (Jiang et al., 2025) focuses on tool-using LLM agents and aligns well with the VeRL codebase. Most prior work provides limited analysis of agentic RL training instability. In contrast, ARLArena offers a unified training and analysis framework for examining how policy-gradient design choices relate to stability

| Dimension | Method | ALFWorld | | WebShop | |
|---|---|---|---|---|---|
| | | Task Score | Success Rate | Task Score | Success Rate |
| Base | GRPO | 2.37 | 50.92 | 85.48 | 73.98 |
| Loss Agg | GRPO$_{ST}$ | 1.68$_{\downarrow 29.1\%}$ | 49.31$_{\downarrow 3.2\%}$ | 91.21$_{\uparrow 6.7\%}$ | 83.57$_{\uparrow 13.0\%}$ |
| Importance Sampling | SAPO | 0.08$_{\downarrow 96.6\%}$ | 1.93$_{\downarrow 96.21\%}$ | 84.73$_{\downarrow 0.9\%}$ | 74.47$_{\uparrow 0.7\%}$ |
| | CISPO | 0.80$_{\downarrow 66.2\%}$ | 30.83$_{\downarrow 39.5\%}$ | 87.80$_{\uparrow 2.7\%}$ | 73.74$_{\downarrow 0.3\%}$ |
| | GSPO | 5.05$_{\uparrow 113.1\%}$ | 79.70$_{\uparrow 56.5\%}$ | 91.61$_{\uparrow 7.2\%}$ | 83.15$_{\uparrow 12.4\%}$ |
| Advantage Design | GIGPO | 4.10$_{\uparrow 73.0\%}$ | 80.03$_{\uparrow 57.2\%}$ | 89.26$_{\uparrow 4.4\%}$ | 78.91$_{\uparrow 6.7\%}$ |
| | EMPG | 4.51$_{\uparrow 90.3\%}$ | 71.48$_{\uparrow 40.4\%}$ | 88.60$_{\uparrow 3.6\%}$ | 75.46$_{\uparrow 2.0\%}$ |
| Dynamic Sampling | DAPO$_{GRPO}$ | 0.81$_{\downarrow 65.8\%}$ | 38.11$_{\downarrow 25.16\%}$ | 86.52$_{\uparrow 1.2\%}$ | 76.52$_{\uparrow 3.4\%}$ |
| | DAPO$_{GIGPO}$ | 2.49$_{\uparrow 5.1\%}$ | 60.27$_{\uparrow 18.4\%}$ | 91.92$_{\uparrow 7.5\%}$ | 82.42$_{\uparrow 11.4\%}$ |
| **Ours** | **SAMPO** | **8.98**$_{\uparrow 278.9\%}$ | **97.71**$_{\uparrow 91.9\%}$ | **93.43**$_{\uparrow 9.3\%}$ | **84.02**$_{\uparrow 13.6\%}$ |

*Table 7.* Performance on SFT version of **Qwen3-8B** for **ALFWorld** and **WebShop** correspondingly. The overall trend on the 8B variant remains consistent, and SAMPO continues to achieve the best performance, indicating stable gains under model scaling.

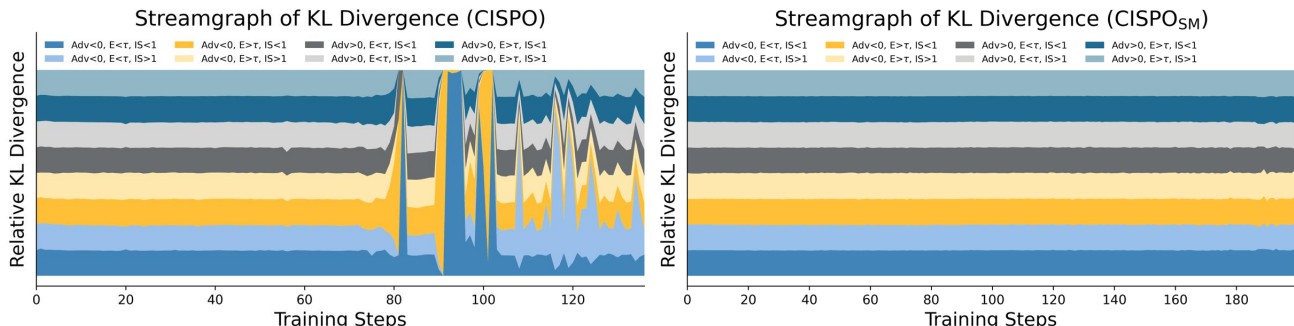

*Figure 4.* Sequence-Level IS Analysis of CISPO and CISPO$_{SM}$ (CISPO with sequence masking) on ALFWorld.

and performance across agentic tasks.

# E. Another Roadmap of Building Agentic LLM: Multi-agent System

## E.1. Debate

Let $\mathbb{A} = \{\mathcal{A}_1, \mathcal{A}_2, \ldots, \mathcal{A}_N\}$ denote the set of $N$ agents, where $N$ is an odd integer to prevent tie-breaking scenarios during majority voting. Let $x$ denote the task prompt. In the initial round ($t = 0$), each agent $\mathcal{A}_i$ independently generates a candidate solution $c_i^{(0)}$ based solely on the prompt $x$:

$$c_i^{(0)} = \mathcal{A}_i(x), \quad \forall i \in \{1, \ldots, N\} \quad (32)$$

Let $\mathcal{C}^{(t)} = \{c_1^{(t)}, c_2^{(t)}, \ldots, c_N^{(t)}\}$ be the set of candidate solutions at round $t$. We define a majority consensus function $\mathcal{M}(\cdot)$ that returns the solution $y$ if it appears in more than half of the agent responses:

$$y = \mathcal{M}(\mathcal{C}^{(t)}) = \begin{cases} \hat{c} & \text{if } \left|\{c \in \mathcal{C}^{(t)} : c = \hat{c}\}\right| > \frac{N}{2} \\ \emptyset & \text{otherwise} \end{cases} \quad (33)$$

If $\mathcal{M}(\mathcal{C}^{(0)}) \neq \emptyset$, the process terminates and outputs $y$. Otherwise, the system enters the debate phase. The process iterates through debate rounds $t = 1, 2, \ldots, T_{max}$. For each round, we construct the debate prompt for each agent, which includes the original prompt $x$, the set of unique candidate solutions from the previous round $\text{Unique}(\mathcal{C}^{(t-1)})$, and agents' reasoning in previous round $\mathcal{R}^{(t-1)}$. Let $\mathcal{R}^{(t)} = \{r_1^{(t)}, r_2^{(t)}, \ldots, r_N^{(t)}\}$ be the agents' reasoning at round $t$, and $\mathcal{R}^{(0)} = \emptyset$.

$$r_i^{(t)}, c_i^{(t)} = \mathcal{A}_i\left(x, \text{Unique}(\mathcal{C}^{(t-1)}), \mathcal{R}^{(t-1)}\right). \quad (34)$$

At the end of each round $t$, we check for consensus again and output the solution $y$ if consensus is reached. This mechanism enables agents to either rectify perceived flaws by proposing a new solution or align with a peer by voting for an existing candidate. The debate terminates when a

majority consensus is achieved, $\mathcal{M}(\mathcal{C}^{(t)}) \neq \emptyset$. If the maximum iteration limit $T_{max}$ is reached without consensus, the final output $y$ is randomly sampled from the final set of candidates $\mathcal{C}^{(T_{max})}$.

## E.2. Aggressive Debate

We extend the Debate framework discussed above to build a decisively goal-oriented variant designed to prioritize task completion over exhaustive exploration. While the standard framework seeks consensus on an optimal solution, the aggressive variant compels agents to accept partial success by securing the best available option within a strict finite horizon.

Formally, we modify the agent $\mathcal{A}_i$ by conditioning it on an additional constraint set $\mathcal{I}_{agg}$. Unlike standard debate agents that aim for a perfect solution, the aggressive agent $\mathcal{A}_i(\cdot | \mathcal{I}_{agg})$ operates under a modified utility function characterized by several governing principles: **(1) Bounded Exploration:** The agent must finalize the interaction within a finite horizon. This constraint suppresses excessive exploration and ensures the agent commits to a definitive outcome rather than prolonging the information-gathering phase; **(2) Temporal Efficiency:** The agent is encouraged to conclude the interaction as early as possible; **(3) Incentive Awareness:** The agent is explicitly informed that partial rewards are available. This awareness incentivizes the agent to accept high-utility suboptimal outcomes when a perfect solution is unattainable; **(4) Pragmatic Optimization:** The agent prioritizes securing a result that maximizes available partial rewards rather than seeking a theoretical global optimum, thereby avoiding diminishing returns associated with perfecting the solution in complex environments.

## E.3. Experiment Results on SLA and MAS

# F. Failure Analysis

## F.1. Method: Sankey Graphs for Action-Transition Flows

We analyze agent rollouts by visualizing step-wise action transitions with Sankey graphs. Each column corresponds to a time step, node height indicates the empirical frequency of an action at that step, and edges represent transitions between consecutive steps. Compared with action histograms, Sankey graphs preserve temporal structure and thus reveal loop-like behaviors (e.g., repetitive pagination or oscillation between two actions) that dominate long-horizon failures.

## F.2. WebShop: Action-Transition Patterns and Failure Modes

**Overall flow (API agent).** The API agent is a single-agent baseline powered by GPT-4o via API under the same interaction protocol, without any task-specific training. Figure 5 summarizes WebShop trajectories of the API agent, where green links correspond to successful episodes and red links correspond to failures. A large fraction of failures is characterized by repetitive `next` actions, suggesting exploration inefficiency where the agent keeps paginating without making progress toward constraint satisfaction.

**Failure-only flow with action coloring.** Figure 6 focuses on failed trajectories and colors nodes by action type. Two dominant failure patterns are observed: (i) **Pagination loops**: long runs of `next` (and occasional `search`) that rarely transition into `click_product` (product-detail inspection); (ii) **Backtracking oscillation**: frequent alternation between `click_product` and `back`, suggesting repeated revisits to previously viewed product pages and limited progress toward constraint satisfaction. Notably, our API agent is provided with a long interaction history (past actions and observations) in the prompt, so this pattern is unlikely to be explained by insufficient context alone. Instead, it may reflect limited *effective* memory usage: without structured tracking or summarization of verified attributes and visited items, the agent may fail to retrieve previously established evidence from a long, unstructured context and thus re-check similar products. We emphasize that this is only one plausible factor; we find instruction ambiguity or conflicting constraints may also contribute.

## F.3. WebShop: How RL Post-training Changes Behaviors

**Overall flow (RL-optimized agent).** Figure 7 shows the same visualization for our RL-optimized agent (post-trained with RL). Compared with the API baseline, the RL agent exhibits fewer `next`-dominated failure paths and a higher proportion of trajectories that transition into `click_product` and eventually attempt `buy`, consistent with more targeted product inspection and earlier decision making.

**Remaining failure modes after RL post-training.** Figure 8 focuses on failed RL trajectories. While `next`-heavy pagination loops become less prominent, two residual issues remain: (i) **Backtracking-heavy browsing**: repeated `click_other`/`back` transitions, suggesting inefficient navigation; (ii) **Premature purchase**: occasional `buy` attempts that do not satisfy all constraints, suggesting incomplete constraint tracking.

| Method | ALFWorld | | | | | | | WebShop | |
|---|---|---|---|---|---|---|---|---|---|
| | Pick | Look | Clean | Heat | Cool | Pick2 | All | Score | Success |
| GPT-4o | 61.11 | 33.33 | 36.36 | 50.00 | 45.45 | 63.64 | 50.00 | 13.60 | 12.50 |
| GPT-5.2 | 70.03 | 66.07 | 35.37 | 62.30 | 52.08 | 37.36 | 51.56 | 26.56 | 26.56 |
| Debate | 67.74 | 64.28 | 33.33 | 60.00 | 52.38 | 65.00 | 56.25 | 34.65 | 22.65 |
| Aggressive Debate | – | – | – | – | – | – | – | 61.53 | 28.51 |
| Gemini-2.5-pro | 84.97 | 61.61 | 63.94 | 22.22 | 62.50 | 75.25 | 66.41 | – | – |
| GRPO | 87.41 | 62.65 | 46.42 | 72.28 | 58.89 | 38.37 | 72.61 | 75.32 | 57.71 |
| SAPO | 34.49 | 32.19 | 24.13 | 24.92 | 16.21 | 9.37 | 25.16 | 73.85 | 52.10 |
| CISPO | 76.03 | 37.12 | 58.56 | 50.97 | 57.88 | 23.68 | 54.42 | 67.96 | 54.71 |
| GSPO | 90.36 | 79.31 | 90.71 | 75.45 | 77.95 | 48.95 | 78.61 | 85.29 | 72.48 |
| GIGPO | 94.80 | 83.03 | 86.37 | 81.15 | 75.38 | 59.21 | 81.09 | 67.76 | 56.55 |
| EMPG | 84.18 | 61.53 | 69.83 | 72.49 | 46.51 | 0.04 | 57.91 | 79.16 | 64.32 |
| DAPO$_{\text{GRPO}}$ | 81.28 | 37.57 | 53.97 | 40.16 | 51.28 | 6.43 | 49.58 | 62.43 | 46.17 |
| DAPO$_{\text{GIGPO}}$ | 85.04 | 55.26 | 65.35 | 58.98 | 56.52 | 26.57 | 60.55 | 88.10 | 76.82 |
| **SAMPO** | **96.30** | **88.49** | **93.65** | **92.42** | **92.70** | **88.35** | **92.72** | **88.04** | **74.08** |

*Table 8.* Unified comparison across ALFWorld (six task types + overall) and WebShop (score and success rate). The upper block reports closed-source baselines and multi-agent strategies; the lower block reports policy optimization methods trained with Qwen3-4B.

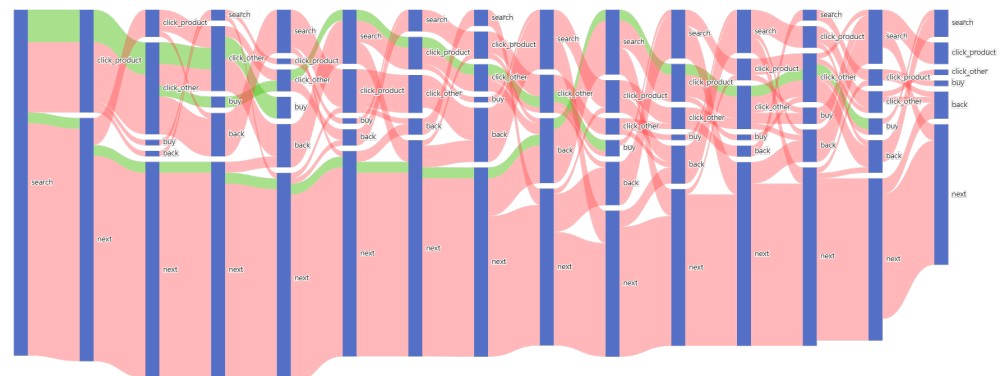

*Figure 5.* WebShop action-transition Sankey for the API agent. Green flows denote successful trajectories and red flows denote failures.

## F.4. ALFWorld: Action-Transition Patterns

Figure 9 visualizes ALFWorld rollouts. Navigation actions (e.g., `go`, `look`) dominate early steps across episodes, whereas successful trajectories more often transition into object-centric interactions (e.g., `examine`, `open/close`, `take`, `use`) and explicit state-checking (`inventory`). In contrast, failed trajectories frequently exhibit prolonged navigation with comparatively fewer object interactions, which may reflect weak progression toward concrete object-level subgoals and imperfect tracking of what has already been tried or collected over long horizons.

## F.5. Implications

Our analysis suggests two actionable directions: (1) **Loop-aware control** (e.g., detecting repeated `next` or `click_product` ↔ `back` cycles and triggering a plan change); (2) **Explicit constraint/state memory** (e.g., introducing a lightweight memory agent that maintains a concise record of visited items and verified constraints, and feeds the acting agent with short summaries or retrieval results). Together, these mechanisms may further improve robustness beyond RL post-training.

## G. Visualization

### G.1. Evidence of Format v.s. Dynamic Filtering

To support the analysis in Section 4.4, we report the format validity ratio during training for different policy optimization variants. The results illustrate that DAPO combined with GIGPO maintains more stable format behavior than DAPO+GRPO after dynamic filtering.

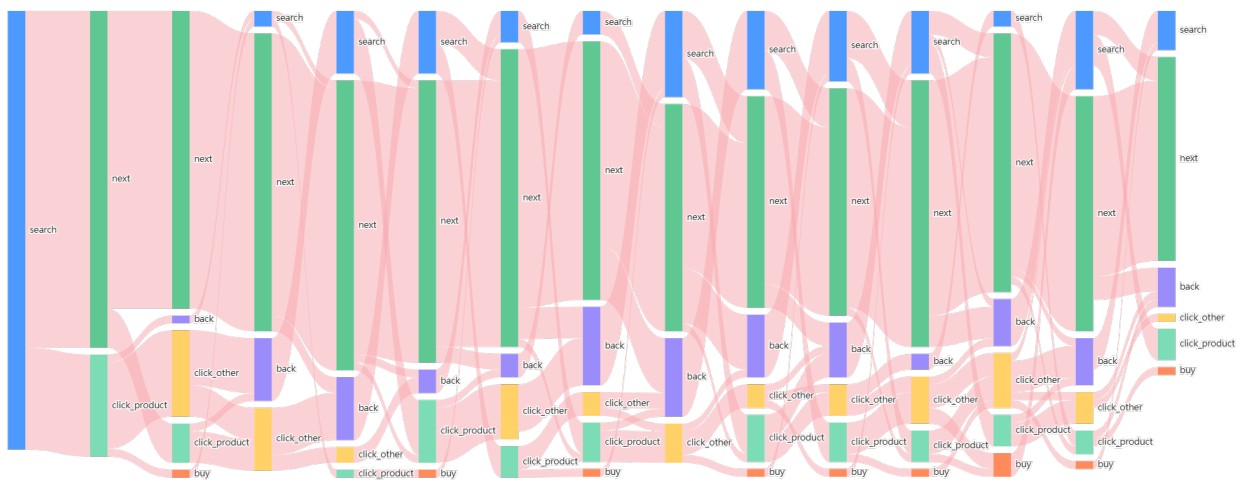

*Figure 6.* WebShop failure-only action-transition Sankey for the API agent. Nodes are colored by action type (e.g., `search`, `click_product`, `click_other`, `buy`, `back`, `next`).

## H. Case Study

### H.1. Prompt Templates

#### H.1.1. TIR MATH

> **TIR Math Prompt**
>
> ```
> TIR_TEMPLATE = """
> Solve the following problem step by step
>     .
> You now have the ability to selectively
>     write executable Python code
> to enhance your reasoning process.
>
> The Python code will be executed by an
>     external sandbox, and the output
> (after "Code execution result: ") is
>     returned to aid your reasoning
> and help you arrive at the final answer.
>
> The Python code should be complete
>     scripts, including necessary imports.
>
> Code Format:
> Each code snippet is wrapped between
>     triple backticks.
> You need to use `print()` to output
>     intermediate results.
>
> Answer Format:
> You can use the `final_answer()`
>     function in the code to return your
> final answer.
>
> For example, to answer the User Question
>     :
> What is the result of the 5 + 3 +
>     1294.678?, you can write:
> ```

```
    answer = 5 + 3 + 1294.678
    final_answer(answer)

You can also use \boxed to return your
    answer.
The last part of your response should be
    :
\boxed{'The final answer goes here.'}

User Question:
"""
```

#### H.1.2. WEBSHOP

> **WebShop Prompt**
>
> ```
> WEBSHOP_TEMPLATE = """
> You are an expert autonomous agent
>     operating in the WebShop
> e-commerce environment.
> Your task is to: {task_description}.
>
> Prior to this step, you have already
>     taken {step_count} step(s).
> Below are the most recent {
>     history_length} observations and the
> corresponding actions you took:
> {action_history}
>
> You are now at step {current_step} and
>     your current observation is:
> {current_observation}.
> Your admissible actions of the current
>     situation are:
> [
> {available_actions}
> ].
> ```

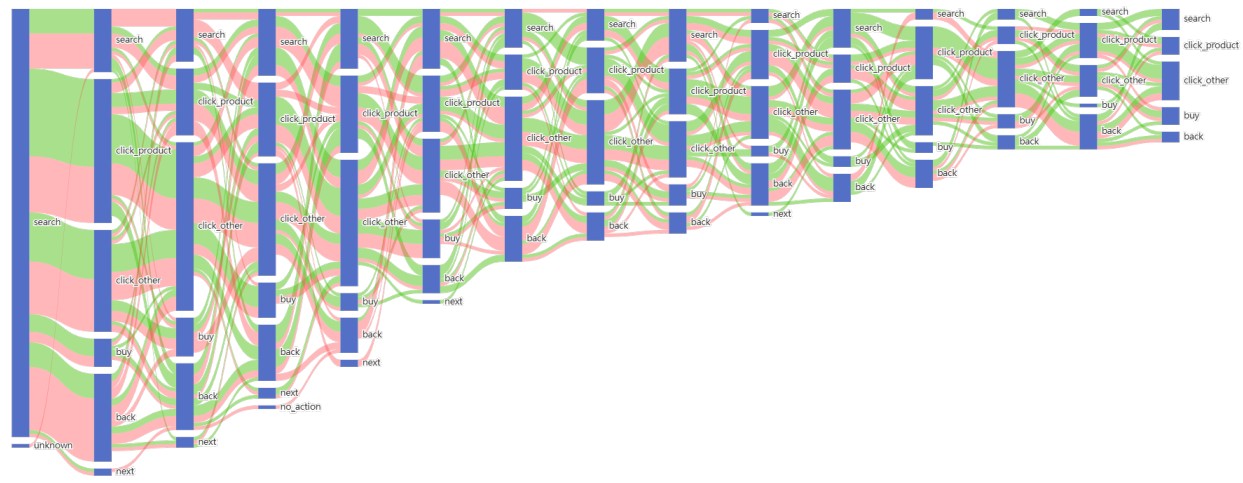

*Figure 7.* WebShop action-transition Sankey for the RL-optimized agent. Green flows denote successful trajectories and red flows denote failures.

```
Now it's your turn to take one action
    for the current step.
You should first reason step-by-step
    about the current situation,
then think carefully which admissible
    action best advances the
shopping goal. This reasoning process
    MUST be enclosed within
<think> </think> tags.
Once you've finished your reasoning, you
     should choose an admissible
action for the current step and present
    it within <action> </action>
tags.
"""
```

```
Your admissible actions of the current
    situation are:
[{admissible_actions}].

Now it's your turn to take an action.
You should first reason step-by-step
    about the current situation.
This reasoning process MUST be enclosed
    within <think> </think> tags.
Once you've finished your reasoning, you
    should choose an admissible
action for the current step and present
    it within <action> </action> tags.
"""
```

### H.1.4. SOKOBAN

**Sokoban Prompt**

```
SOKOBAN_TEMPLATE = """
You are an expert agent operating in the
    Sokoban environment.
Your goal is to push all the boxes onto
    the target spots.
Once all boxes are on the targets, you
    win!

# Rules
You can only push boxes. You can't pull
    them, so plan ahead to
avoid getting stuck.
You can't walk through or push boxes
    into walls.
To avoid traps, do not push boxes into
    corners or against walls
where they can't be moved again.
```

### H.1.3. ALFWORLD

**ALFWorld Prompt**

```
ALFWORLD_TEMPLATE = """
You are an expert agent operating in the
     ALFRED Embodied Environment.
Your task is to: {task_description}

Prior to this step, you have already
    taken {step_count} step(s).
Below are the most recent {
    history_length} observations and the
corresponding actions you took:
{action_history}

You are now at step {current_step} and
    your current observation is:
{current_observation}
```

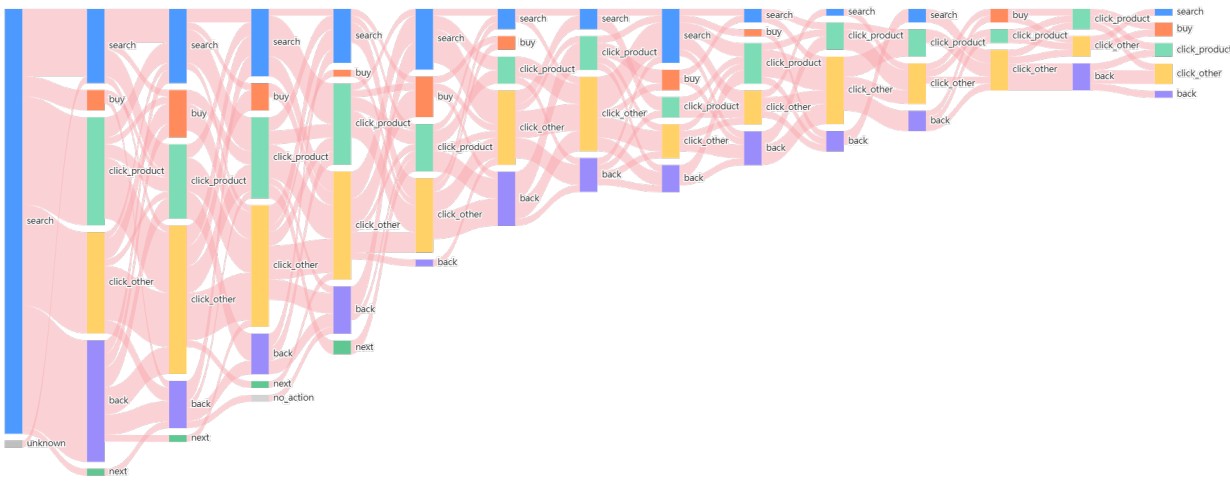

*Figure 8.* WebShop failure-only action-transition Sankey for the RL-optimized agent. Nodes are colored by action type (e.g., `search`, `click_product`, `click_other`, `buy`, `back`, `next`).

```
# Visual Elements in the Image:
Character: A small, green alien-like
    figure with two antennae and
black eyes. It represents you.
Box: A yellow crate marked with an
    orange "X" across its front.
It is the box you need to push.
Target: A black tile outlined in red,
    with a small red diamond
shape in the center. It marks the
    destination where a box should
be pushed.

# Current Step
Your current observation is shown in the
     image: <image>
Your admissible actions are ["up", "down
    ", "left", "right"].

Now it's your turn to make a move (
    choose ONE action only for the
current step).
You should first reason step-by-step
    about the current situation -
observe the positions of boxes and
    targets, plan a path to push a
box toward a target, and avoid traps
    like corners or walls.
This reasoning process MUST be enclosed
    within <think> </think> tags.
Once you've finished your reasoning, you
     should choose an admissible
action for the current step and present
    it within <action> </action> tags.
"""
```

## H.2. Multi-turn State-Action Templates

### H.2.1. TIR MATH

> **TIR Math Rollout Example**
>
> **Input:**
> ```
> Let a, b, and c be real numbers such
>     that a >= b >= c >= 0
> and a + b = 16. Find the maximum value
>     of
>      sqrt(c(b - c)) + sqrt(c(a - c)).
> ```
>
> **Output:**
> To solve this problem step by step:
>
> 1. Define the constraints and variables.
>
> 2. Use Python to iterate through possible values of $a$, $b$, and $c$.
>
> 3. Calculate the expression for each set of values.
>
> 4. Keep track of the maximum value found during iterations.
>
> 5. Return the maximum value.
>
> Let's implement this approach:
>
> ```python
> import numpy as np
>
> # Define the range and step size for
>     a and b since we know they sum
>     to 16
> a_vals = np.arange(0, 17, 0.01)
> b_vals = np.arange(0, 17, 0.01)
>
> max_value = -np.inf
>
> ```

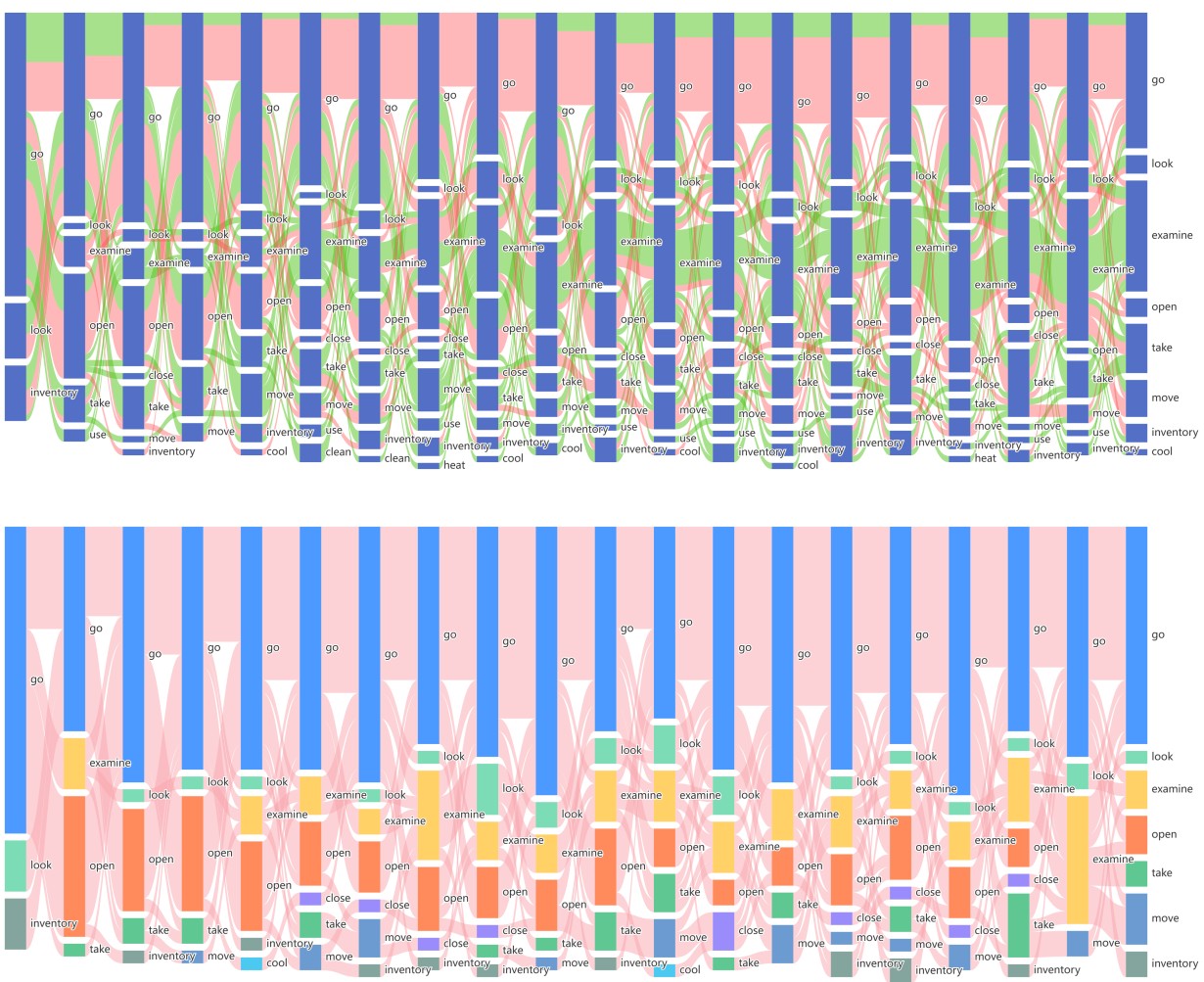

*Figure 9.* ALFWorld action-transition Sankey diagrams for the API agent. Top: Success (green) vs. failure (red) trajectories. Bottom: Failure trajectories with nodes colored by action type.

```python
 9   # Iterate through possible values of
        a and b
10   for a in a_vals:
11       for b in b_vals:
12           if a + b == 16 and a >= b:
13               # Iterate through
        possible values of c
14               for c in np.arange(0,
        min(b, a), 0.01):
15                   # Check if this
        configuration satisfies all
        constraints
16                   if c <= b and c <= a
        and a >= b >= c:
17                       # Calculate the
        expression
18                       expr_val = np.
        sqrt(c * (b - c)) + np.sqrt(c *
        (a - c))
```

```python
19                       # Update the
        maximum value if necessary
20                       if expr_val >
        max_value:
21                           max_value =
        expr_val
22
23   # Print the maximum value found
24   print(f"Maximum Value: {max_value}")
```

*Code execution result:* Maximum Value:  8.0

8

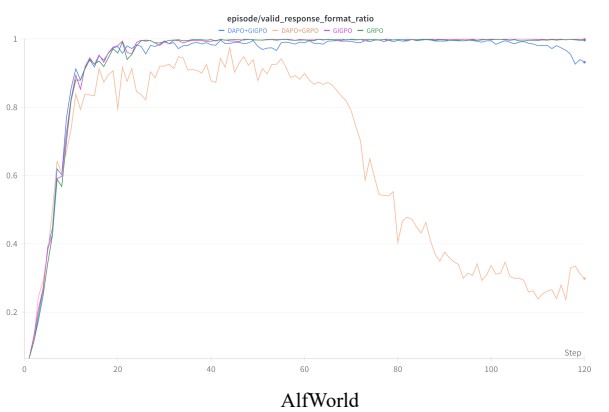 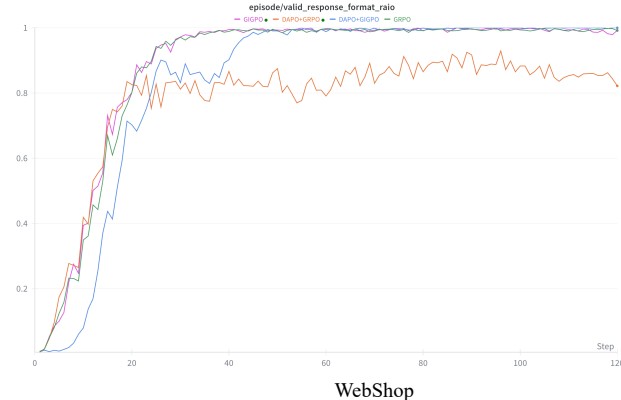

AlfWorld                                         WebShop

*Figure 10.* Format validity ratio during training on AlfWorld and WebShop for GRPO, GIGPO, DAPO$_{\text{GRPO}}$, and DAPO$_{\text{GIGPO}}$. Applying dynamic filtering to GRPO leads to degraded format stability, whereas DAPO$_{\text{GIGPO}}$ maintains stable format behavior across training.

### H.2.2. WEBSHOP

**WebShop Rollout Example (one turn)**

**Input:**

```
user

You are an expert autonomous agent
    operating in the WebShop e-commerce
environment.
Your task is to: Find me machine
    washable men's t-shirts with short
    sleeve
for tumble dry with color: 36 pack mix,
    and size: xx-large, and price lower
than 160.00 dollars.

Your current observation is: 'Search'.

Your admissible actions of the current
    situation are:
[
'search[<your query>]',
'click[search]',
].

Now it's your turn to take one action
    for the current step.
You should first reason step-by-step
    about the current situation, then
    think
carefully which admissible action best
    advances the shopping goal. This
reasoning process MUST be enclosed
    within <think> </think> tags.

Once you've finished your reasoning, you
    should choose an admissible action
for the current step and present it
    within <action> </action> tags.
```

**Output:**

```
<think>
Okay, so I need to find machine washable
    men's t-shirts, short sleeve,
for tumble dry, 36 pack mix, xx-large,
    under $160.
The current observation is 'Search', so
    I need to create a specific query
that includes all these parameters.
</think>

<action>
search[men's short sleeve t-shirts, 36
    pack mix, xx-large, machine washable
    ,
tumble dryable, color: 36 pack mix,
    price: $160 or less]
</action>
```

### H.2.3. ALFWORLD

**ALFWorld Rollout Example (one turn)**

**Input:**

```
user

You are an expert agent operating in the
    ALFRED Embodied Environment.
Your current observation is: -= Welcome
    to TextWorld, ALFRED! =-

You are in the middle of a room. Looking
    quickly around you, you see a
cabinet 13, a cabinet 12, a cabinet 11,
    a cabinet 10, a cabinet 9,
a cabinet 8, a cabinet 7, a cabinet 6, a
    cabinet 5, a cabinet 4,
a cabinet 3, a cabinet 2, a cabinet 1, a
    coffeemachine 1, a countertop 1,
a diningtable 1, a drawer 1, a fridge 1,
    a garbagecan 1, a microwave 1,
```

```
a shelf 3, a shelf 2, a shelf 1, a
    sinkbasin 1, a stoveburner 4,
a stoveburner 3, a stoveburner 2, a
    stoveburner 1, and a toaster 1.

Your task is to: put a cool egg in
    microwave.

Your admissible actions of the current
    situation are:
[
'go to cabinet 1', ..., 'go to cabinet
    13',
'go to coffeemachine 1', 'go to
    countertop 1', 'go to diningtable
    1',
'go to drawer 1', 'go to fridge 1', 'go
    to garbagecan 1',
'go to microwave 1', 'go to shelf 1', '
    go to shelf 2', 'go to shelf 3',
'go to sinkbasin 1', 'go to stoveburner
    1', ..., 'go to stoveburner 4',
'go to toaster 1', 'inventory', 'look'
].

Now it's your turn to take an action.
You should first reason step-by-step
    about the current situation.
This reasoning process MUST be enclosed
    within <think> </think> tags.
Once you've finished your reasoning, you
     should choose an admissible action
for the current step and present it
    within <action> </action> tags.
```

**Output:**
```
<think>
Okay, I need to put a cool egg in the
    microwave.
First, where would I find a cool egg?
    The fridge is typically where you
keep cool items like eggs. So maybe I
    should start by going to the fridge.

Looking at the list of admissible
    actions, there's "go to fridge 1".
That's probably the first step. Once
    there, I can check the interior of
the fridge to see if there's an egg.

Alternatively, maybe I should check the
    dining table or the countertops
in case the egg is already on a surface.
     But I think the fridge is more
likely to have an egg. Since the task is
     to put a cool egg in the
microwave, it's probably in the fridge.
    So I should go to the fridge first.
</think>

<action>
go to fridge 1
```

```
</action>
```

### H.2.4. SOKOBAN

**Sokoban Rollout Example (one turn)**

**Input:**
```
user

You are an expert agent operating in the
     Sokoban environment.
Your goal is to push all the boxes onto
    the target spots.
Once all boxes are on the targets, you
    win!

# Rules
You can only push boxes. You can't pull
    them, so plan ahead to
avoid getting stuck.
You can't walk through or push boxes
    into walls.
To avoid traps, do not push boxes into
    corners or against walls where
they can't be moved again.

# Visual Elements in the Image:
Character: A small, green alien-like
    figure with two antennae and
black eyes. It represents you.
Box: A yellow crate marked with an
    orange "X" across its front.
Target: A black tile outlined in red,
    with a small red diamond shape
in the center.

# Current Step
Your current observation is shown in the
     image.
Your admissible actions are ["up", "down
    ", "left", "right"].

Now it's your turn to make a move (
    choose ONE action only for the
current step).
You should first reason step-by-step
    about the current situation –
observe the positions of boxes and
    targets, plan a path to push a box
toward a target, and avoid traps like
    corners or walls.
This reasoning process MUST be enclosed
    within <think> </think> tags.
Once you've finished your reasoning, you
    should choose an admissible
action for the current step and present
    it within <action> </action> tags.
```

**Output:**
```
<think>
```

```
To push the box to the target, I need to
    move it to the right.
The current position is a corner, which
    is a trap.
I should avoid pushing the box into
    corners or against walls.
The path is straightforward: move right.
This action will allow me to push the
    box towards the target without
getting stuck.
</think>

<action>
right
</action>
```

