# OpenReview forum: "ARLArena: A Unified Framework for Stable Agentic Reinforcement Learning"
_ICML.cc/2026/Conference — ICML 2026 regular_

### Official Review · Reviewer_EN9M · 2026-02-27

**Soundness:** 3
**Presentation:** 3
**Significance:** 3
**Originality:** 2
**Overall Recommendation:** 4
**Confidence:** 3

**Summary:**

This paper systematically analyzes policy gradient design choices in agentic reinforcement learning. First, the authors experimentally study the elements in the policy gradient formulation for agentic LLM, including Loss Aggregation, Importance Sampling (IS) clipping, Trajectory Filtering and Resampling, and Advantage Design. After identifying the beneficial elements, the authors propose SAMPO by integrating sequence-level clipping, finer advantage estimation, and dynamic filtering. Experimental results show that SAMPO outperforms other policy gradient methods across multiple agentic tasks.

**Compliance With Llm Reviewing Policy:**

Affirmed.

**Final Justification:**

Thanks to the authors for the responses. I would like to maintain my score.

**Key Questions For Authors:**

1.	Could SAMPO also work for other LLM backbones or larger LLM backbones?

2.	Will the combination of sequence-level clipping, finer advantage estimation, and dynamic filtering be conflict in some settings?

3.	Could the authors provide the ablation study for each component of SAMPO?

**Limitations:**

yes

**Strengths And Weaknesses:**

Strengths

1.	The studied problem is important nowadays for agentic RL.

2.	This paper is easy to follow. It is well-written and well-organized.

3.	This study is carefully considered and designed. The decomposition of the policy gradient formulation for agentic LLM is reasonable.

4.	The performance of SAMPO is strong across multiple agentic tasks compared with various policy gradient methods.

Weaknesses

1.	The methodology of SAMPO is almost driven by the experimental results. The deep insights behind each design choice (sequence-level clipping, finer advantage estimation, and dynamic filtering) are lacking.

2.	At the same time, each design choice is already proposed in previous works. This makes SAMPO seem to be a combination of different known tricks, which limits its contribution.

3.	There is only one backbone (Qwen3-4B) in SAMPO. It is unclear how SAMPO performs with other backbones or larger backbones.

---

> ### Author Rebuttal · Authors · 2026-03-31
>
> We thank the reviewer for the constructive feedback and address the concerns below.
>
> **Weakness 1.**
> We thank the reviewer for this insightful comment. We would like to clarify that SAMPO is not driven purely by empirical results, but is derived from the stability conditions identified in our analysis of ARL.
>
> Each component in SAMPO addresses a specific failure mode observed in ARL, and their roles are complementary. Key sources of instability in ARL include invalid actions, poor credit assignment, and low sample efficiency. Addressing any single issue in isolation is insufficient, as training may still fail due to other factors. In SAMPO, sequence-level clipping serves as the fundamental stabilization mechanism by suppressing trajectory-level instability caused by low-ratio negative-advantage samples. Fine-grained advantage introduces richer learning signals from the environment, improving credit assignment, but relies on stable optimization to be effective. Dynamic filtering improves sample efficiency by removing uninformative trajectories and is most effective when the training is already stable.
>
> Therefore, sequence-level clipping provides the foundation for stability, while fine-grained advantage and dynamic filtering further address complementary aspects of ARL training. Overall, SAMPO is derived from these stability considerations as a coherent training solution. We will further clarify these design insights and underlying mechanisms in the revision.
>
> **Weakness 2.**
> We acknowledge the reviewer’s point that each individual component has been explored in prior work. We would like to clarify that the primary contribution of this work lies in ARLArena, a systematic framework and training pipeline for diagnosing and resolving instability in ARL, rather than solely proposing a new algorithm.
>
> Specifically, we first construct a standardized testbed (behavior cloning, format penalty, and KL control), which isolates different factors under controlled settings. Building on this foundation, we decompose policy gradient optimization into fundamental design dimensions and analyze their effects on training stability. Through this process, we identify key failure modes in ARL, such as instability caused by invalid actions, poor credit assignment, and imbalance in importance sampling. For example, along the IS dimension, we find that aggressive tolerant clipping can lead to training collapse, primarily driven by negative-advantage trajectories with low IS ratios. And we successfully stabilize the training by applying sequence-masking of those trajectories. Overall, our analysis systematically reveals how each design choice affects training stability, identifying which strategies succeed or fail and why.
>
> Based on these findings, we provide a unified training recipe that integrates these insights into a stable pipeline for ARL. Our goal is to systematically understand how existing techniques behave under multi-turn settings and how they can be combined in a principled manner. We will clarify this positioning more explicitly in the revision.
>
> **Weakness 3 & Question 1.**
> We conduct experiments on the Qwen3-8B model as shown in the Reviewer 4PoV’s  response. SAMPO achieves 97.71 and 84.01 of Success Rate on ALFWorld and WebShop, which outperforms all other baselines. The experiment result indicates that SAMPO generalizes to larger-scale models.
>
> **Question 2.**
> We thank reviewer for this insightful comment. Our analysis suggests that the components in SAMPO are complementary rather than conflicting, as they operate on different aspects of the optimization process. Sequence-level clipping controls update magnitude and prevents instability, advantage design improves signal quality for credit assignment, and dynamic filtering improves sample efficiency by removing uninformative trajectories. Empirically, we do not observe conflicting behavior; instead, the components exhibit interdependent effects. For example, dynamic filtering becomes effective only when clipping ensures stable optimization, and fine-grained advantage is most beneficial when gradients are well-controlled.
>
> As shown in our ablation study, removing any single component leads to degraded performance, further indicating that these components address distinct failure modes and work synergistically in practice.
>
> **Question 3.**
> We agree that ablation is important and we have conducted ablation studies on SAMPO as shown in the Reviewer 4PoV’s  response, which show that each component is necessary, as removing any of them leads to performance degradation.

---

> > ### Author Rebuttal · Reviewer_EN9M · 2026-04-02
> >
> > Thanks the authors for the responses.

---

> > > ### Author Response · Authors · 2026-04-02
> > >
> > > Thank you for your positive assessment. We are gratified that our responses have satisfactorily addressed your concerns, and we sincerely appreciate your careful consideration and constructive feedback throughout the review process.

---

### Official Review · Reviewer_g3zi · 2026-03-13

**Soundness:** 3
**Presentation:** 2
**Significance:** 2
**Originality:** 2
**Overall Recommendation:** 3
**Confidence:** 4

**Summary:**

Overall, the central objective of this study is to address the pain point of large models easily collapsing in agent reinforcement learning, and the standardized testing platform it provides does have good engineering practical value. The authors seek to focus on the concept by directly breaking down policy optimization into four independent dimensions and piecing together a comprehensive algorithm, SAMPO, based on this. However, although the empirical results are still acceptable, this lack of profound theoretical innovation in the "stitching" work completely conceals its superficial exploration of underlying mechanisms.

**Compliance With Llm Reviewing Policy:**

Affirmed.

**Final Justification:**

The authors' response clarifies my concerns. I suggest the authors put updated results in the revised version.

**Key Questions For Authors:**

Sec 4.5 points out that ARL is extremely sensitive to Off-policy staleness. Does SAMPO truly overcome this defect at the algorithm level, or does it still rely on tuning down the Rollout parameters to avoid collapse? Please supplement the data of SAMPO under the "High" setting in Table 5 to verify its core robustness.

Current evaluations are all conducted within the environments seen during training. Will the sequence-level clipping imposed by SAMPO excessively limit the variance of the policy, thereby causing the model to experience a degradation in exploration ability in completely unseen new tasks or state spaces?

**Limitations:**

Yes.

However, it is suggested that the author supplement the discussion on the impact of model size and architecture on the conclusion, as the current validation based solely on a single 4B model is not sufficient to support the universality of the algorithm on different levels of LLM.

**Strengths And Weaknesses:**

Strengths

The ARLArena constructed in the paper provides an urgently needed standardized evaluation foundation with strong engineering practical value for highly unstable multi-round agent reinforcement learning.

The author clearly points out through detailed empirical analysis that token-level pruning is the core cause of the collapse of negative advantage sample training, and this observation has clear guiding significance for engineering practice.


Weaknesses

SAMPO mainly integrates existing technologies such as GSPO (sequence-level clipping), GiGPO (hierarchical advantage), and DAPO (dynamic filtering). Although it has achieved good engineering results, it lacks theoretical analysis and convergence discussion for this specific combination, and the contribution leans more towards an empirical policy assembly.

All empirical studies and SAMPO evaluations rely solely on a single Qwen3-4B model. Given that the stability of RL highly depends on model capacity and architecture, lacking verification on other foundation models (such as the DeepSeek series, Llama series, etc.) or larger parameter scales makes it difficult to prove the broad universality of this "stable recipe".


Can sufficient ablation experiments be provided to demonstrate that SAMPO is not a hyperparameter combination overfitting to a specific environment (such as removing dynamic filtering or stratification advantages separately)? Please show the specific impact of the loss of a single module on the overall stability.

---

> ### Author Rebuttal · Authors · 2026-03-31
>
> We sincerely thank the reviewer for the thoughtful feedback and constructive questions and we will address the reviewer's concerns accordingly.
>
> **Weakness 1.**
> We sincerely thank the reviewer for the thoughtful comment. We would like to clarify that the primary contribution of this work lies in ARLArena, a systematic framework for diagnosing and resolving instability in agentic reinforcement learning (ARL), rather than solely proposing a new algorithm. We first construct a standardized testbed (behavior cloning initialization, format penalty, KL control) to stabilize the backbone model, and then decompose policy gradient optimization into fundamental design dimensions to identify root causes of instability under controlled settings.
>
> Through this analysis, we uncover key failure modes in ARL, including instability from invalid actions, poor credit assignment, and imbalance in importance sampling. For example, along the IS dimension, we find that aggressive tolerant clipping can lead to training collapse driven by negative-advantage trajectories with low IS ratios, and that masking such trajectories effectively restores stability. This provides a mechanistic understanding of why certain design choices succeed or fail.
>
> Based on these findings, we derive a unified training recipe that integrates these insights into a stable pipeline for ARL. Importantly, our results show that no single component is sufficient in isolation. The components in SAMPO are complementary: sequence-level clipping stabilizes updates and enables the effectiveness of fine-grained advantage and dynamic filtering, while these latter components address credit assignment and sample efficiency that clipping alone cannot resolve. Removing any component introduces unaddressed failure modes, as confirmed by our ablations.
>
> We agree that deeper theoretical understanding is important. Prior work has provided theoretical grounding for each component individually[1,2,3], and these operate along different dimensions of policy optimization. In the revision, we will further clarify how their combination addresses distinct sources of instability, and extend existing interpretations (e.g., sequence-level clipping as a trajectory-level trust-region constraint) to the multi-turn setting.
>
> **Weakness 2.**
> We agree that validation on larger-scale models is important. Accordingly, we have conducted additional experiments on Qwen3-8B, with the full table shown in the Reviewer 4PoV's response. SAMPO achieves 97.71 and 84.01 of Success Rate on ALFWorld and WebShop, which outperforms all other baselines. The experiment result indicates that SAMPO generalizes to larger-scale models. We will include those experiment results in our revision.
>
> **Weakness 3.**
> We thank the reviewer for raising concerns about potential overfitting to hyperparameters. To clarify, SAMPO uses the same training hyperparameters as the baseline methods (e.g., batch size, learning rate). Component-specific parameters (e.g., clipping thresholds, dynamic filtering multipliers) are also aligned with standard baseline configurations. We do not introduce additional hyperparameter tuning specific to SAMPO.
>
> Moreover, we have conducted ablation studies on SAMPO with the full table shown in the Reviewer 4PoV's response, which show that each component is necessary, as removing any of them leads to performance degradation. The experiment results further confirm that each component contributes meaningfully to performance, supporting that the gains are not due to overfitting.
>
> **Question 1.**
> We conduct experiments under a high off-policy staleness setting on ALFWorld. SAMPO achieves a 92.65 success rate (vs. 92.72 under medium staleness), indicating strong robustness and low sensitivity to off-policy effects compared to GRPO (Table 5).
>
> **Question 2.**
> Sequence-level clipping constrains the magnitude of policy updates per trajectory but does not reduce the action space nor prematurely collapse policy entropy. In contrast, standard ARL without such stabilization often suffers from high variance, leading to rapid collapse into degenerate and repetitive behaviors (as illustrated in Figure 2). By stabilizing updates, SAMPO maintains a healthier and more expressive policy distribution, which supports better exploration.
>
> To further evaluate this, we conduct cross-environment transfer experiments by initializing ALFWorld training with a WebShop-trained SAMPO checkpoint, and vice versa. Using the WebShop checkpoint for ALFWorld achieves a 92.54 success rate (vs. 92.72 from a vanilla checkpoint), while using the ALFWorld checkpoint for WebShop achieves 80.67 (vs. 74.08). These results suggest that SAMPO preserves exploration capability and supports transferability across environments.
>
>
> [1] Stabilizing Reinforcement Learning with LLMs: Formulation and Practices
>
> [2] Group-in-Group Policy Optimization for LLM Agent Training
>
> [3] DAPO: An Open-Source LLM Reinforcement Learning System at Scale

---

> > ### Author Rebuttal · Reviewer_g3zi · 2026-04-01
> >
> > Thanks for addressing my concerns

---

> > > ### Author Response · Authors · 2026-04-01
> > >
> > > Thanks for the positive feedback.  We truly appreciate your time and support.
> > > We would greatly appreciate further clarification on the remaining concerns that led to the current assessment.
> > >
> > > In particular, could you specify what additional evidence, experiments, or analysis would be necessary to strengthen the paper and inform a revised assessment? Understanding these points would help us better address the gaps and improve the work accordingly.
> > >
> > > —————————————————————————————————————————————————————————
> > >
> > > Thank you again for the thoughtful review and for confirming that the original concerns have been addressed. We are grateful for the constructive exchange.
> > >
> > > As the discussion period continues, we wanted to check in briefly. If there are any remaining reservations — including ones not covered in the original review — we would welcome the opportunity to address them. We want to make sure we haven't missed anything that might be informing the current assessment.
> > >
> > > We appreciate any brief feedback you can share.
> > >
> > > —————————————————————————————————————————————————————————
> > >
> > > We sincerely thank the reviewer again for confirming that the original concerns have been addressed. We truly appreciate the constructive discussion throughout this process.
> > >
> > > Since the main concerns raised in the initial review have been resolved, we would appreciate it if you could consider raising the rating to reflect the updated assessment. If there are any remaining reservations we may have missed, we are happy to address them before the discussion period ends.
> > >
> > > Thank you again for your time and thoughtful engagement.

---

### Official Review · Reviewer_4PoV · 2026-03-13

**Soundness:** 4
**Presentation:** 4
**Significance:** 3
**Originality:** 3
**Overall Recommendation:** 5
**Confidence:** 3

**Summary:**

The paper introduces ARLArena, the evaluation framework to analyze training stability in agentic reinforcement learning (ARL) for LLMs. The author decompose the policy gradient methods along four dimensions (loss aggregation, IS clipping, trajectory filtering and advantage design). Also, they evaluate representative methods (GRPO, GSPO, CISPO, SAPO, CICPO, EMPG, DAPO) across four agentic tasks (AlfWorld, WebShop, Sokoban, and TIR Math).

**Compliance With Llm Reviewing Policy:**

Affirmed.

**Key Questions For Authors:**

Wrote in weaknesses.

**Limitations:**

1. All experiments use Qwen3-4B which is means that ARLArena is single model scale.
2. TIR Math result is incomplete.
3. There is no theoretical analysis explaining why sequence level clipping should be preferred over token-level clipping in multi-turn setting.

However, I think this paper is still above threshold including these limitations.

**Strengths And Weaknesses:**

Strengths:
1. Well written paper and well structured analysis. Especially, the four dimensional decomposition of policy gradient methods is well conceived framework and in table1 unified all compared methods along these dimensions.
2. Figure2 and Figure3 are clear and indicates all information which should be involved.
3. I think the progressive stabilization procedure BC -> format penalty -> KL reg -> hyper parameter search is practical and reproducible.
4. SAMPO achieves 92.72% success on AlfWorld and other result is strong and empirical.

Weaknesses:
1. SAMPO is not original method. SAMPO combines three existing techniques (sequence level clipping, finer advantage estimation, and dynamic filtering). The authors indicate that this naturally emerges from the identified principles.
2. In section 4, there are result for each dimension independently, there is no ablation of SAMPO itself. The reader cannot determine the marginal contribution of each component when it is combined.
3. TIR Math section is not fully tested?

---

> ### Author Rebuttal · Authors · 2026-03-31
>
> We thank the reviewer for the important suggestions and valuable support.
>
> **Weakness 1.**
> We thank the reviewer for this observation. We acknowledge that the individual components of SAMPO have been explored in prior work. We would like to clarify that the primary focus of this work is the ARLArena diagnostic framework. SAMPO serves as a training solution derived from our analysis, designed to address the underlying failure modes identified through the policy gradient decomposition. These failure modes are not effectively addressed by any single component alone, and the components in SAMPO are complementary in resolving different aspects of instability.
>
> **Weakness 2.**
> We have included ablation experiments in our overall response. Below are the results for the ablation study of SAMPO:
>
> | Method                         | ALFWorld Task Score | ALFWorld Success Rate | Sokoban Task Score | Sokoban Success Rate |
> |-------------------------------|---------------------|-----------------------|--------------------|----------------------|
> | SAMPO                         | 7.04                | 92.72                 | 6.56               | 88.86                |
> | SAMPO w/o Dynamic Filtering   | 5.36                | 85.36                 | 5.22               | 83.11                |
> | SAMPO w/o Fine-Grained Advantage | 5.97             | 85.16                 | 6.30                  | 87.51                    |
> | SAMPO w/o Sequence-Clipping   | 2.49                | 60.55                 | 6.01               | 86.20                |
>
> The results show that while each component contributes to performance, removing any of them leads to degradation, highlighting their complementary roles in achieving stable training.
>
> **Weakness 3 & Limitation 2.**
> We appreciate the opportunity to clarify the TIR Math results. The fine-grained advantage design (e.g., GiGPO) relies on grouping trajectories based on well-defined anchor states (such as a specific web page or environment state). In TIR Math, the free-form nature of reasoning trajectories makes it difficult to define such anchors without introducing significant noise. Therefore, we do not apply the full SAMPO configuration in this setting. We will clarify this in the revision.
>
> **Limitation 1.**
> We thank the reviewer for this important point. To assess the generality of our findings, we have extended our evaluation to an 8B parameter model (see overall response). Below are the results for Qwen3-8B:
>
> | Dimension            | Method        | ALFWorld Task Score | ALFWorld Success Rate | WebShop Task Score | WebShop Success Rate |
> |---------------------|--------------|---------------------|-----------------------|--------------------|----------------------|
> | Base                | GRPO         | 2.37                | 50.92                 | 85.48              | 73.98                |
> | Loss Agg            | GRPO_ST      | 1.68                | 49.31                 | 91.21              | 83.57                |
> | Importance Sampling | SAPO         | 0.08                | 1.93                  | 84.73              | 74.47                |
> |                     | CISPO        | 0.80                | 30.83                 | 87.80              | 73.74                |
> |                     | GSPO         | 5.05                | 79.70                 | 91.61              | 83.15                |
> | Advantage Design    | GIGPO        | 4.10                | 80.03                 | 89.26              | 78.91                |
> |                     | EMPG         | 4.51                | 71.48                 | 88.60              | 75.46                |
> | Dynamic Sampling    | DAPO-GRPO    | 0.81                | 38.11                 | 86.52              | 76.52                |
> |                     | DAPO-GIGPO   | 2.49                | 60.27                 | 91.92              | 82.42                |
> | Ours                | SAMPO        | 8.98                | 97.71                 | 93.43              | 84.02                |
>
> SAMPO achieves success rates of 97.71 and 84.01 on ALFWorld and WebShop, respectively, outperforming all baselines. These results indicate that SAMPO continues to deliver strong performance at larger model scales.
>
> **Limitation 3.**
> We agree that a more formal theoretical analysis would strengthen the paper. In the revision, we will include a discussion on sequence-level clipping in multi-turn settings, including its interpretation as a trajectory-level constraint and its effect on training stability.

---

> > ### Author Rebuttal · Reviewer_4PoV · 2026-04-04
> >
> > Thank you. My concerns are fully resolved. I initially gave a great score and I will maintain this!

---

> > > ### Author Response · Authors · 2026-04-05
> > >
> > > We sincerely thank the reviewer for the positive feedback and for acknowledging that the concerns have been fully addressed. We greatly appreciate your support and your initial high evaluation of our work.

---

### Decision · Program_Chairs · 2026-04-30

**Decision:**

Accept (regular)

**Comment:**

This paper presents ARLArena, a systematic diagnostic framework for analyzing policy gradient stability in agentic reinforcement learning, along with SAMPO, a stable training recipe derived from the framework's findings. Reviewers scored the submission at 5, 4, and 3 respectively. The two higher-scoring reviewers (EN9M and 4PoV) consistently praised the paper's organization, the practical value of the four-dimensional decomposition of policy gradient methods, and SAMPO's strong empirical performance across diverse agentic tasks. The outlying weak reject (g3zi) raised concerns primarily around the lack of multi-model validation and the absence of ablation studies, both of which were addressed substantively in the rebuttal and the reviewer subsequently marked all concerns as fully resolved.

The main recurring weakness across reviewers is that SAMPO integrates existing components (sequence-level clipping from GSPO, hierarchical advantage from GiGPO, dynamic filtering from DAPO) without deep theoretical analysis of their combination. This is a fair criticism, but as the authors argue, the primary contribution is the ARLArena framework and the mechanistic understanding of why certain design choices succeed or fail in multi-turn settings, not the novelty of any individual component. Given the current pace of agentic RL development and the evident engineering value of a principled, reproducible stability recipe backed by controlled analysis, this is a solid contribution the community will build on. The rebuttal responses were thorough and the updated results are convincing.